# Adaptive Fission: Post-training Encoding for Low-latency Spike Neural Networks

**Yizhou Jiang**[1]    **Feng Chen**[1]    **Yihan Li**[1]    **Yuqian Liu**[1]

**Haichuan Gao**[2]    **Tianren Zhang**[1*]  **Ying Fang**[3*]

[1]Department of Automation, Tsinghua University, Beijing, China
[2]Beijing QianJue Technology Co., Ltd.
[3]College of Computer and Cyber Security, Fujian Normal University, Fuzhou, China
{jiangyz20, lyh19, liuyuqian21}@mails.tsinghua.edu.cn
haichuan.gao@qj-robots.com, chenfeng@mail.tsinghua.edu.cn
trzhang@mail.tsinghua.edu.cn, fy20@fjnu.edu.cn

## Abstract

Spiking Neural Networks (SNNs) often rely on rate coding, where high-precision inference depends on long time-steps, leading to significant latency and energy cost—especially for ANN-to-SNN conversions. To address this, we propose Adaptive Fission, a post-training encoding technique that selectively splits high-sensitivity neurons into groups with varying scales and weights. This enables neuron-specific, on-demand precision and threshold allocation while introducing minimal spatial overhead. As a generalized form of population coding, it seamlessly applies to a wide range of pretrained SNN architectures without requiring additional training or fine-tuning. Experiments on neuromorphic hardware demonstrate up to 80% reductions in latency and power consumption without degrading accuracy.

## 1 Introduction

Unlike traditional artificial neural networks (ANNs) that use floating-point activations, spiking neural networks (SNNs) encode activations as binary spike sequences over time, promising energy-efficient intelligence on neuromorphic hardware [31, 5]. However, they still lag behind ANNs in precision and performance due to inefficient rate-based encoding. While a $T$-bit quantized ANN represents $2^T$ distinct values, an SNN needs $T$ time-steps with 1-bit outputs per step to encode only $T+1$ discrete states, as illustrated in Fig. 1. This makes high-precision inference slow and energy-intensive on complex tasks such as generation [44, 20].

Inspired by training-aware quantization [9, 51], recent efforts aim to train SNNs from scratch with variable bit-lengths or time-steps [45, 37], thereby reducing reliance on temporal precision. However, in many real-world cases such as open-vocabulary detection, SNNs are converted from large, pretrained ANNs (e.g., applications of CLIP [35, 24, 50, 48]), where retraining is infeasible due to proprietary data or computational cost. To address this, existing post-training methods either modify spike bits [42, 41] or increase firing steps locally [27], but yield only modest gains. Moreover, most neuromorphic chips lack support for multi-bit, multi-threshold neurons or dynamic time-steps. Therefore, a hardware-friendly post-training solution for low-latency deployment remains critically needed.

---

[*]Corresponding authors.
[1]Code is available at: https://github.com/JiangYizhou16/Adaptive-Fission

39th Conference on Neural Information Processing Systems (NeurIPS 2025).

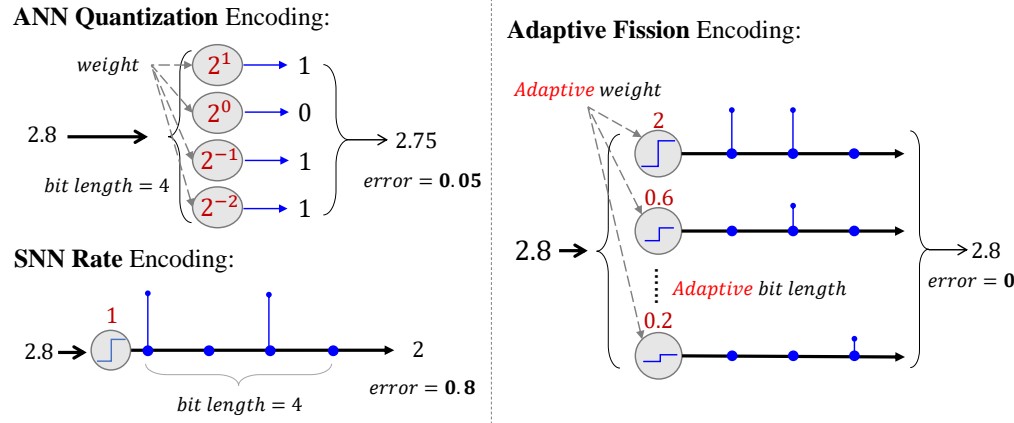

Figure 1: An overview of encoding methods. *Adaptive Fission* combines the sparsity of SNN rate encoding with the efficiency of ANN quantization, enabling variable bit-lengths and weights.

As a classical yet underutilized alternative, population coding distributes representation across groups of neurons, replacing temporal integration with spatial parallelism and thus reducing latency. Prior works [40, 30, 24] typically adopt static group structures with fixed scales, weights, and thresholds, resulting in substantial memory overhead and hindering practical deployment. In contrast, we propose Adaptive Fission, a flexible, generalized population-coding strategy that selectively splits high-sensitivity neurons into dynamically sized groups and allocates precision and thresholds on demand. This neuron-wise bit allocation enables mixed-precision inference while remaining compatible with binary neuron models. Our analysis shows it approaches the theoretical capacity of population coding with only moderate spatial overhead.

To implement Adaptive Fission, we design a two-stage iterative pipeline to address two key challenges: identifying which neurons should undergo fission and determining the scales and thresholds within each neuron group. In the first stage, we estimate and rank neuron sensitivity by jointly analyzing gradients and residual membrane potentials, prioritizing those with the greatest impact. This targeted reallocation achieves capacity comparable to fixed-length population encoding with only $20\%$ additional neurons. In the second stage, we derive an optimality condition for assigning thresholds to the newly created sub-neurons, based on the cumulative distribution of membrane potentials. Although analytically intractable, we approximate the solution using Monte Carlo sampling combined with Newton's method, yielding minimal encoding error.

By enhancing representational precision under strict latency constraints, Adaptive Fission enables faster, more energy-efficient inference. As a post-training technique, it is compatible with both ANN-to-SNN conversions and directly trained SNNs, and can be combined with quantization and pruning to offset the modest increase in neuron count. Deployments on a Lynxi HP201 neuromorphic chip [34] demonstrate up to $80\%$ reductions in latency and energy consumption without compromising accuracy. Beyond classification, Adaptive Fission also supports generative tasks like image synthesis, highlighting its potential for high-throughput neuromorphic computing.

Our key contributions are as follows:

1. We propose Adaptive Fission, a post-training population-coding method that assigns variable bit lengths and weights, with theoretical guarantees of exponential error reduction.
2. We introduce a practical, hardware-aware encoding pipeline for parallel SNN acceleration, compatible with both ANN-to-SNN conversion and directly trained SNNs.
3. Evaluation on a neuromorphic platform demonstrates up to $80\%$ reductions in latency and power consumption while maintaining competitive accuracy.

## 2 Related Work

### 2.1 Spike Encoding

Rate coding [1, 11] remains the dominant strategy in SNNs, where all spikes contribute equally with temporal information discarded. Despite its simplicity and robustness, it suffers from low information density. Variants such as signed or ternary spikes [42, 14] and variable spike amplitudes

or thresholds [15, 47, 8, 41] enhance expressivity but often compromise hardware compatibility. Alternatively, population coding leverages multiple neurons to jointly represent scalar values. While promising, existing works typically employ fixed-scale ensembles [45, 37] or rely on hand-crafted heuristics for neuron substitution [24], limiting precision scalability. Besides, temporal coding [12, 39, 32, 22] uses spike timing rather than frequency to improve encoding density, but it currently underperforms rate coding. Our method can be viewed as a generalized framework of population coding with adaptive, hardware-aware support.

## 2.2 Latency Reduction in SNNs

Reducing the number of simulation steps is critical for latency and energy-efficiency. Existing efforts can be categorized into three main types: **(1) Post-training conversions** introduce burst spikes [29] or negative spikes [42], but still typically require $> 32$ time-steps to match ANN-level accuracy. **(2) Quantization-aware conversions** emulate low-bit activations via modified activation functions [17, 4], or employ pre-charged neurons [23, 3] to reduce inter-layer delays, bringing time-steps down to 8–16 but necessitating training from scratch. **(3) Direct training approaches** treat SNNs as recurrent networks and use surrogate gradients [33, 52] or synaptic plasticity [25, 2], with recent work incorporating variable-bit activations [43, 37]. Despite achieving 2–8 time-steps, they suffer from training instability and scalability issues in larger networks [46]. Our method complements these approaches by offering a post-training optimization without retraining or architectural modifications, and is fully compatible with both converted and directly trained SNNs.

## 3 Preliminaries

As our method operates at the neuron level, we first formulate the model from the perspective of individual neurons.

**ANN models:** In a standard feedforward ANN with ReLU activation, each neuron receives input from presynaptic activations $a_{pre}^j$ via synaptic weights $w^j$. The total input $q_{ann}$ and activation $a$ are:

$$q_{ann} = \sum_j w^j a_{pre}^j, \quad a = \max\{q_{ann}, 0\}. \tag{1}$$

**SNN models with weighted spikes:** We focus on soft-reset Integrate-and-Fire (IF) neurons [36, 16] and adopt the weighted-spike formulation commonly used in ANN–SNN conversion [28]. Let $\theta$ be the firing threshold, and $s_{pre}^j(t)$ the presynaptic spike at time-step $t$. The total input is $q_{snn}(t) = \sum_j w^j s_{pre}^j(t)$, and membrane dynamics are defined as:

$$\text{Charging: } v_{temp}(t) = v(t-1) + q_{snn}(t), \tag{2}$$

$$\text{Firing: } s(t) = \theta \cdot H\big(v_{temp}(t) - \theta\big), \tag{3}$$

$$v(t) = v_{temp}(t) - s(t), \tag{4}$$

where $v(t)$ is the membrane potential after step $t$, and $H(\cdot)$ the Heaviside function. Here $s(t)$ is not binary $\{0, 1\}$ but weighted $\{0, \theta\}$. To support binary-only hardware, we absorb $\theta$ into synaptic weights:

$$w := \theta w, \quad s(t) := s(t)/\theta = H\big(v_{temp}(t) - \theta\big) \in \{0, 1\}. \tag{5}$$

For simplicity, we continue using the $\{0, \theta\}$ notation, where $\theta$ also denotes the effective output weight.

**SNN and Quantized ANNs:** In rate coding, a neuron's activation intensity is represented by its average firing rate over $T$ time-steps, i.e., $s = \frac{\sum_{t=1}^T s(t)}{T}$. Assuming $\sum_t q_{snn}(t) = q_{ann} = a$, the SNN activation approximates a $T$-level quantized ANN:

$$s = \frac{a - v(T)}{T} = \frac{\theta \cdot clip\left(\left\lfloor \frac{a}{\theta} \right\rfloor, 0, T\right)}{T}. \tag{6}$$

Thus, the firing rate approximates the quantized ANN activation, with quantization error stemming from $\theta$ and reflected in the residual potential $v(T)$. Increasing $T$ distributes activation across more steps, allowing a smaller threshold $\theta \approx \max(a)/T$ for finer quantization.

# 4 Population Encoding Models

Most SNNs assign a single binary neuron to each activation, limiting precision under temporal constraints. We propose a generalized population coding framework where multiple binary neurons jointly encode a single activation, enabling finer quantization while preserving hardware-friendly binarization. Unlike prior methods that require retraining or weight updates, our approach is entirely post-training and model-agnostic, making it applicable to both converted and directly trained SNNs.

## 4.1 Static Models and Error Analysis

To understand the precision limits of multi-neuron representations, we first isolate static encoding behavior, independent of spike timing.

**Definition 4.1** (Encoding Error). Consider a group of $k$ neurons with distinct thresholds $\theta_i \in \mathbb{R}^+$. Let $n_i$ denote the spike count of neuron $i$ within $T$ time-steps. The set of representable values is:

$$\mathcal{S} = \left\{ 0, \sum_{i=1}^{k} \theta_i n_i + b \,\middle|\, n_i \in \{0, 1, \dots, T\} \right\}, \tag{7}$$

where $b$ is constant bias. Given an input distribution $P(q)$ with $q \geq 0$, the expected encoding error is:

$$r = \mathbb{E}_q \left[ \min_{s \in \mathcal{S}, s \leq q} (q - s) \right]. \tag{8}$$

In this context, a conventional IF neuron is a simplified case with $k = 1$. For a fixed $\theta$, the optimal firing count is $n = clip\left( \lfloor \frac{q-b}{\theta} \rfloor, 0, T \right)$. Assuming a uniform feature distribution, the expected encoding error can be directly computed, aligning with previous results [7].

**Proposition 4.2** (Optimal Error of Single Neuron). *If $q \sim U(0, q_{\max})$, the minimal error of a single neuron is given by $r = \frac{q_{\max}}{2(T+1)}$ with the threshold and bias $\theta = 2r$ and $b = r$.*

Extending to multi-neuron encoding (Def. 4.1), the absence of fixed firing patterns across neurons prevents direct computation of individual spike counts $n_i$. Yet under the uniform assumption, an optimal configuration can still be derived without explicit firing rules.

**Proposition 4.3** (Optimal Error of Neuron Groups). *For a $k$-neuron group encoding $q \sim U(0, q_{\max})$, the optimal error $r$, threshold $\theta_i$ and bias $b$ are given by:*

$$r = \frac{q_{\max}}{2(T+1)^k}, \; \theta_i = \frac{q_{\max}}{(T+1)^i}, \; b = r. \tag{9}$$

As the number of neurons increases, the error decreases exponentially, yielding far substantial precision gains compared to the linear improvement from increasing time-steps $T$. This result, however, captures only aggregate encoding after all $T$ steps, providing an optimal hindsight solution while ignoring spike dynamics.

## 4.2 Temporal Dynamic Encoding Models

Previous encoding analyses typically assume full knowledge of the input $q$ over all time-steps. In contrast, practical SNN inference operates under an online regime, where input is incrementally accumulated and spikes are emitted as soon as the membrane potential crosses the threshold. This scenario can be viewed as a special case of Proposition 4.3 with $T = 1$.

**Proposition 4.4** (Optimal Error with Temporal Dynamics). *For a $k$-neuron group, assume the residual membrane potential at the final step follows $P\big(v_{temp}(T)\big) \in U(0, \frac{q_{\max}}{T})$. The optimal error $r$, threshold $\theta_i$ and bias $b$ are given by*

$$r = \frac{q_{\max}}{T \cdot 2^{k+1}}, \theta_i = \frac{q_{\max}}{T \cdot 2^i}, b = r. \tag{10}$$

This proposition implies that exponentially increasing precision ($2^{k+1}$ levels) can still be achieved under online temporal encoding by assigning thresholds as powers of two. Moreover, it provides

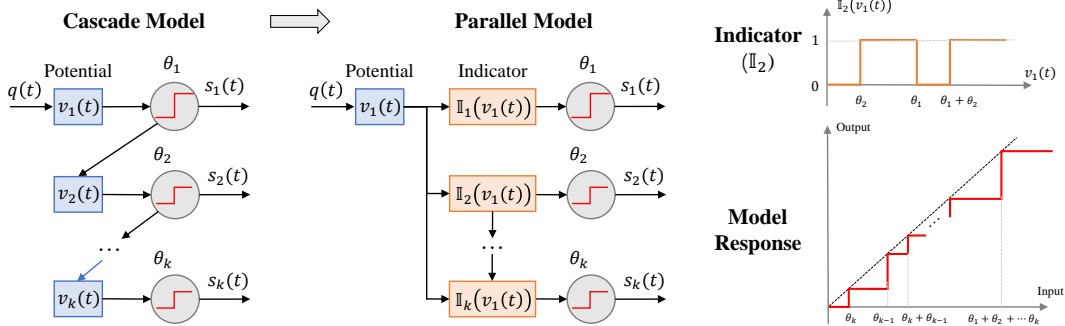

Figure 2: Dynamics of our proposed neuron groups. Neurons are arranged in a hierarchical structure based on descending thresholds, and can be parallelized by pre-compiled indicators. Their response of input spikes forms a multi-level step function as an uneven quantization.

theoretical support for recent neuron-grouping heuristics [30]. However, these results critically rely on the assumption of a uniformly distributed residual potential, which rarely holds in practice, where long-tailed or multi-modal distributions are more common.

To generalize the analysis to arbitrary input distributions, a more explicit mathematical formulation of a neuron group's temporal dynamics is necessary. While multiple formulations are possible, we adopt a descending-threshold population model (Fig. 2), where the residual input is successively propagated to neurons with lower thresholds after higher-threshold neurons fire. Although this formulation is inherently sequential, it admits an equivalent parallel implementation that preserves efficiency.

**Definition 4.5** (Spiking Neuron Group Model). Consider a group of $k$ neurons with descending thresholds $\theta_1 \geq \theta_2 \geq \cdots \geq \theta_k$. At time-step $t$, let $q(t)$ denote the total synaptic input, $v_i(t)$ the residual potential of neuron $i$, and $s_i(t)$ its weighted output spike.

Initialize the first neuron's potential as $v_1(1) = q(1)$. Then, for all $t = 1, \ldots, T$ and $i = 1, \ldots, k$:

$$\text{Firing:} \qquad s_i(t) = \theta_i \cdot H\left(v_i(t) - \theta_i\right), \tag{11}$$

$$\text{Transfer:} \qquad v_{i+1}(t) = v_i(t) - s_i(t), \tag{12}$$

$$\text{Charging:} \qquad v_1(t+1) = v_k(t) - s_k(t) + q(t+1). \tag{13}$$

The total output $s(t)$ is the sum of individual weighted spikes:

$$s(t) = \sum_{i=1}^{k} s_i(t). \tag{14}$$

Although the above dynamics is presented as a sequential cascade—where higher-threshold neurons fire first and residual potential is propagated downward—the computation can equivalently be realized in a fully parallel manner. Specifically, each neuron's firing condition depends solely on the total input and the predefined threshold set, eliminating the need for explicit synchronization with upstream firing events. This property enables the cascade to be "precompiled" into lightweight parallel comparison logic. Formally, once the thresholds $\theta_i$ are fixed, all spike firings can be expressed as binary indicator functions $\mathbb{I}_i$ composed of Heaviside terms:

$$s_i(t) := \theta_i \cdot \mathbb{I}_i(v_1(t)) = \theta_i \cdot \sum_{m_k \in \{0,1\}} \left\{ H\left(v_1(t) - \sum_{j=1}^{i-1} m_j \theta_j\right) - H\left(v_1(t) - \sum_{j=1}^{i-1} m_j \theta_j - \theta_i\right) \right\} \tag{15}$$

This representation requires only comparisons against fixed thresholds and thus introduces negligible overhead on neuromorphic hardware. The resulting hierarchical model forms the foundation for adaptive neuron encoding. By tuning the number of neurons $k$ and the threshold set $\theta_i$ for each activation, precision can be flexibly allocated without retraining or modifying weight values. In the following section, we demonstrate how to determine the optimal configuration of this structure for each neuron based on post-training statistics.

# 5 Methodology: Adaptive Fission Encoding

## 5.1 Overview

In practical networks, individual neurons exhibit diverse activation distributions and precision requirements, making it intractable to derive optimal bit-lengths or threshold assignments analytically.

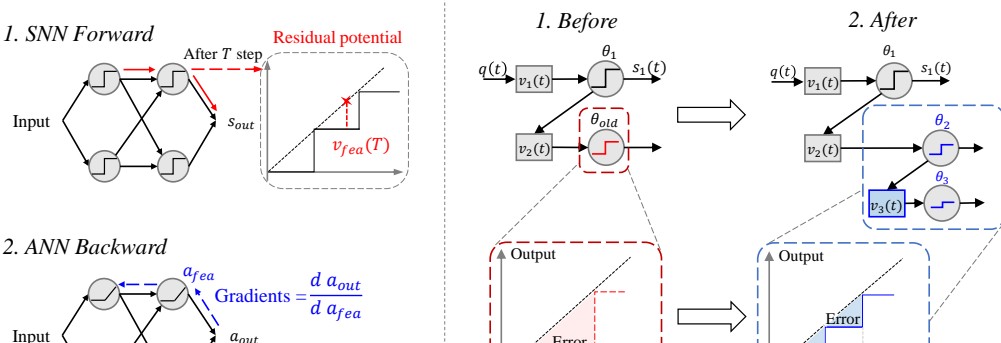

Figure 3: Overview of the two iterative stages in Fission Encoding pipeline. Sensitivity Estimation identifies neurons with high impact using residual potentials and gradients. Threshold Fission splits selected neurons and optimizes their thresholds based on their activation distribution. The process progressively enhances precision while preserving binary spike constraints.

To address this, we adopt a two-stage iterative pipeline, termed Adaptive Fission, as illustrated in Fig.3. Analogous to post-training calibration in ANN quantization [21], the first stage employs a small calibration set to estimate each neuron's precision demand based on its contribution to the output error. This calibration procedure is entirely training-free, as it involves no weight updates and only collects activation statistics, thereby requiring a small number of samples and incurring negligible computational overhead. In the second stage, we selectively allocate additional neurons and adjust their thresholds to minimize the expected residual potential. The algorithm progressively splits the most influential neurons into two units with reduced weights in each iteration. This neuron "fission" mechanism achieves a principled balance between precision and resource consumption while maintaining the original network's accuracy.

## 5.2 Stage 1: Sensitivity Estimation and Selection

To prevent excessive scale increase, we consider the inherent redundancy in deep networks, and focus precision enhancement on neurons with the most significant impact on output accuracy, introducing a "sensitivity" metric to quantify such impact. Specifically, its estimation involves two propagation passes. In the first, a standard SNN inference records the residual potential $v_{fea}(T)$ after $T$ steps. In the second pass, we temporarily substitute spiking neurons with ReLUs to enable ANN-like gradient backpropagation. Denoting the continuous activation of a neuron as $a_{\text{fea}}$, the sensitivity $\sigma_{\text{fea}}$ can be approximated by a first-order Taylor expansion:

$$\sigma_{fea} = \mathbb{E}\|a_{out}(a_{fea}) - a_{out}(a_{fea} - v(T))\|_2 \approx \mathbb{E}\|\frac{da_{out}}{da_{fea}} \cdot v_{fea}(T)\|_2, \qquad (16)$$

where the expectation is taken over calibration samples. Intuitively, this measures how much the residual potential at each activation neuron affects the final output. Neurons are ranked by $\sigma_{fea}$, and only a top fraction (fission rate) are selected for further enhancement.

## 5.3 Stage 2: Optimal Threshold Fission

For each neuron group selected for fission, we add one extra neuron to redistribute thresholds and achieve finer-grained representation. Unlike prior analysis, we estimate the potential distribution directly from calibration data. Because residual potential is mainly determined by the smallest threshold, fission is applied to the last neuron in the group with threshold $\theta_{old}$. We split it into two with thresholds $\theta_k > \theta_{k+1}$ such that $\theta_k + \theta_{k+1} = \theta_{old}$, preserving total contribution. Given the residual potential $v = v_k(T)$, the improved quantization reduces residual error as:

$$\Delta v = (\theta_k - \theta_{k+1})H(v - \theta_k) + \theta_{k+1}H(v - \theta_{k+1}), \qquad (17)$$

and the objective is therefore to determine $\theta_k$ that maximizes the expected error reduction, $\mathbb{E}[\Delta v]$.

Since $\Delta v$ is non-differentiable, closed-form solutions are infeasible. Instead, we derive the optimal thresholds using the cumulative distribution of potential.

**Theorem 5.1** (Optimal Fission Threshold). *Let $F(\cdot)$ denote the cumulative distribution function of residual potential $v$. The optimal threshold $\theta_k$ is given by solution of equation:*

$$2F(\theta_k) - F(\theta_{old} - \theta_k) - 1 = 0, \tag{18}$$

*and the maximum expected reduction in residual potential is:*

$$\mathbb{E}[\Delta v] = \theta_{old} \cdot (1 - F(\theta_k)), \tag{19}$$

*which is bounded by:*

$$\frac{1}{2} \leq \frac{\mathbb{E}[\Delta v]}{\mathbb{E}[v]} \leq 1 \tag{20}$$

*Remark* 5.2. Detailed proof is in the Appendix.A.3 . We first substituting the expectation of $\Delta v$ with an integral over the range of $v$, then transforming it into the cumulative distribution function $F(v)$:

$$
\begin{aligned}
\mathbb{E}[\Delta v] &= \int_{\theta_k}^{\theta_{old}} (\theta_k - \theta_{k+1}) p(v) dv + \int_{\theta_{k+1}}^{\theta_{old}} \theta_{k+1} p(v) dv \\
&= \theta_k \left[ 1 - F(\theta_k) \right] - (\theta_{old} - \theta_k) \cdot \left[ F(\theta_k) - F(\theta_{old} - \theta_k) \right].
\end{aligned}
$$

The extremum of $\theta_k = \arg\max_{\theta_k} \mathbb{E}[\Delta v]$ can be obtained by differentiation. As $F(v)$ is estimated from discrete samples, its derivative $p(v)$ can be neglected, yielding:

$$\frac{d\mathbb{E}[\Delta v]}{d\theta} = [2F(\theta_k) - F(\theta_{old} - \theta_k) - 1] = 0, \tag{21}$$

which provides the optimal solutions in Eq.18.

This result implies that each additional neuron introduced by fission encoding can reduce the quantization error by at least half, even under the worst-case residual distribution. Consequently, the precision exhibits exponential scaling with a base greater than 2, surpassing the efficiency of uniform quantization and demonstrating even stronger empirical performance.

For practical numerical implementation, the search for $\theta_k$ is conducted using a Monte Carlo estimation of $F(v)$ collected from the calibration set. Owing to the piecewise nature of $F(v)$, we first initialize $\theta_k$ using a coarse grid search, followed by refinement via the Newton iteration method:

$$\theta_k := \theta_k - \frac{2F(\theta_k) - F(\theta_{k+1}) - 1}{2p(\theta_k) + p(\theta_{k+1})}, \tag{22}$$

where $p(\theta_k)$ is estimated by sampling within small intervals $\delta$ around $\theta_k$, i.e.,

$$F(\theta)' = p(\theta) \approx \frac{1}{2\delta} \sum_v I(\theta - \delta \leq v \leq \theta + \delta). \tag{23}$$

To further reduce quantization bias caused by Heaviside activation, we adopt a rounding strategy similar to [4] for the last neuron in each group. Specifically, we apply a linear compensation to the input and modify the step function to be symmetric around the threshold:

$$q(t) \leftarrow q(t) + \frac{0.5}{T}\theta_{k+1}, \tag{24}$$

thereby compensating for the flooring bias in spike generation.

## 6 Experiments

### 6.1 Implementations

Fission Encoding is a post-training strategy broadly compatible with diverse SNN architectures. It leverages high-accuracy training with long time-steps while enabling low-latency inference at deployment. We evaluate it across representative paradigms, including continuous-activation conversion Calib [28], quantized conversion QCFS, SRP [4, 17], direct training for spike CNNs TEBN [10], post-training conversion of ANN Transformers STA [24], and spike-based Transformers Spike-driven [49].

For classification, we conduct experiments on CIFAR-100 [26] and ImageNet [6] using ResNet20 [18], VGG16 [38], and a spike-based Transformer [49]. Converted models use $T = 32$ time-steps, while

Table 1: Performance comparison before/after Fission. Mem., Time and Energy per epoch are measured directly on-chip. All accuracy values reflect the best performance at evaluated time-step.

| Method | Time Step | Fission Rate | Accu. (%) | Mem. (GB) | Time/ Epoch(s) | Energy (Wh) |
|---|---|---|---|---|---|---|
| **CIFAR-100 & ResNet20** | | | | | | |
| Calib. (Conversion) | 32 | No | *76.32* | 0.85 | 56.8 | 0.64 |
| | 16 | | 64.48 | | 29.1 | 0.37 |
| | 8 | | 32.10 | | 16.3 | 0.22 |
| | 4 | | 20.43 | | 10.0 | 0.14 |
| **+ Fission** | 16 | 0.10 | 76.20 | 0.88 | 31.4 | 0.42 |
| | 8 | 0.78 | 73.28 | 1.17 | 19.5 | 0.27 |
| | 4 | 1.35 | 73.11 | 1.31 | 12.0 | 0.18 |
| | 2 | 3.70 | 70.50 | 2.25 | **6.4** | **0.11** |
| QCFS. (Quantization) | 32 | No | *76.78* | 0.81 | 59.6 | 0.70 |
| | 16 | | 67.54 | | 30.2 | 0.36 |
| | 8 | | 62.60 | | 16.2 | 0.21 |
| | 4 | | 55.25 | | 9.7 | 0.12 |
| **+ Fission** | 16 | 0.16 | *74.61* | 0.84 | 31.8 | 0.39 |
| | 8 | 0.69 | 72.79 | 1.09 | 16.4 | 0.22 |
| | 4 | 1.17 | 72.15 | 1.22 | 10.1 | 0.18 |
| | 2 | 2.93 | 69.43 | 1.83 | **5.9** | **0.13** |
| SRP. $\tau = 4$ (Quantization) | 32 | No | 65.50 | 0.80 | 55.1 | 0.57 |
| | 16 | | 64.71 | | 28.3 | 0.35 |
| | 8 | | 62.94 | | 15.7 | 0.18 |
| | 4 | | 59.34 | | 8.0 | 0.11 |
| **+ Fission** | 16 | 0.28 | **68.72** | 0.86 | 27.8 | 0.39 |
| | 8 | 1.02 | 66.41 | 1.13 | 14.4 | 0.22 |
| | 4 | 2.71 | 65.39 | 1.65 | 9.1 | 0.16 |
| TEBN. (Training) | 4 | No | *76.13* | 0.83 | 9.8 | 0.18 |
| | 2 | | 56.04 | | 5.0 | 0.11 |
| **+ Fission** | 2 | 1.47 | 73.46 | 1.18 | 5.3 | 0.14 |
| | 1 | 1.95 | 69.20 | 1.50 | **3.2** | **0.09** |
| **CIFAR-100 & ViT-B/32 (ANN Transformer Conversion)** | | | | | | |
| STA. (Conversion) | 64 | No | **85.25** | 8.71 | 128.3 | 1.72 |
| | 32 | | 84.15 | | 70.5 | 1.03 |
| **+ Fission** | 16 | 0.53 | 82.70 | 10.4 | 39.2 | 0.77 |
| | 8 | 1.15 | 80.28 | 15.7 | 24.3 | 0.42 |

| Method | Time Step | Fission Rate | Accu. (%) | Mem. (GB) | Time/ Epoch(s) | Energy (Wh) |
|---|---|---|---|---|---|---|
| **ImageNet & VGG16** | | | | | | |
| Calib. (Conversion) | 32 | No | *63.64* | 3.71 | 604 | 7.55 |
| | 16 | | 55.79 | | 302 | 4.30 |
| | 8 | | 28.10 | | 154 | 2.37 |
| **+ Fission** | 16 | 0.15 | 61.94 | 3.93 | 324 | 4.75 |
| | 8 | 0.87 | 60.79 | 5.28 | 169 | 2.61 |
| | 4 | 1.57 | 58.31 | 6.21 | **88** | **1.43** |
| QCFS. (Quantization) | 32 | No | *68.10* | 3.54 | 627 | 7.82 |
| | 16 | | 50.97 | | 324 | 4.49 |
| **+ Fission** | 16 | 0.23 | 67.32 | 3.87 | 332 | 4.61 |
| | 8 | 0.91 | 64.51 | 5.01 | 157 | 2.91 |
| | 4 | 1.80 | 62.87 | 6.92 | **93** | **1.53** |
| TEBN. | 4 | No | *69.03* | 3.62 | 78 | 1.16 |
| **+ Fission** | 2 | 0.39 | 67.85 | 4.19 | 47 | 0.84 |
| | 1 | 1.31 | 64.19 | 5.77 | **28** | **0.69** |
| **ImageNet & ResNet34** | | | | | | |
| Calib. (Conversion) | 32 | No | 64.82 | 2.68 | 348 | 3.02 |
| | 16 | | 56.21 | | 189 | 1.73 |
| | 8 | | 31.75 | | 110 | 0.95 |
| **+ Fission** | 16 | 0.17 | **66.94** | 2.81 | 214 | 2.01 |
| | 8 | 0.96 | 63.75 | 4.60 | 135 | 1.38 |
| | 4 | 1.72 | 60.43 | 5.57 | **55.2** | **0.61** |
| QCFS. (Quantization) | 32 | No | **69.37** | 2.48 | 372 | 3.29 |
| | 16 | | 59.35 | | 228 | 2.08 |
| **+ Fission** | 16 | 0.28 | 67.93 | 2.96 | 240 | 2.41 |
| | 8 | 1.04 | 66.51 | 4.38 | 156 | 1.54 |
| | 4 | 1.95 | 62.41 | 5.05 | **81.3** | **0.95** |
| **ImageNet & Spike-driven Transformer 8-384** | | | | | | |
| Spike-driven. | 4 | No | *72.28* | 4.57 | 192 | 3.49 |
| **+ Fission** | 2 | 0.54 | 72.41 | 5.54 | 98 | 1.91 |
| | 1 | 0.98 | 70.29 | 6.49 | **60** | **1.02** |

directly trained models use $T = 4$. Fission Encoding is applied post-training with 2,000 calibration samples for CIFAR-100 and 20,000 for ImageNet. For image generation on CIFAR-10, we convert a 4-layer spike CNN generator (WGAN-GP [13]) and a U-Net diffusion model [19], both configured with $T = 64$ and calibrated on the full 60,000-sample dataset. The output layer remains in non-spiking RGB format. All models are deployed on a Lynxi HP201 neuromorphic accelerator (16GB), a commercial successor to Tianjic [34], for real-time measurement of memory usage, latency, and energy consumption. More details see Sec. E.

## 6.2 Main Results on Image Classification

Table 1 summarizes classification results with Fission Encoding across various time-steps, training strategies, and architectures. We report top-1 accuracy, on-chip memory, inference latency per epoch, and energy usage, all measured on the Lynxi HP201 accelerator with a batch size of 32. Fission Encoding consistently improves the accuracy–efficiency trade-off. By reallocating precision spatially instead of extending spike windows temporally, it reduces inference time-steps by up to $8\times$ while maintaining competitive accuracy. For example, decreasing $T$ from 32 to 4 results in less than a $5\%$ accuracy drop, up to $83\%$ energy savings, and only a $1.6\times$ memory increase.

This trade-off is particularly advantageous for neuromorphic SNN deployments, where memory budgets are typically modest. The additional memory introduced by fission remains well below the 16 GB on-chip capacity of the HP201 accelerator, obviating the need for off-chip storage or compres-

sion. Furthermore, the increased neuron count has negligible impact on per-cycle power, while the substantial reduction in time-steps markedly lowers both latency and total energy consumption.

Figure 4 illustrates how accuracy and power consumption scale with time-steps on ResNet20. At moderate $T$, only a small fraction of fission neurons is required to recover full accuracy. Under ultra-low latency constraints ($T = 1$–$2$), additional neurons significantly improve precision, maintaining competitive performance with minimal spatial overhead. These results demonstrate that Fission Encoding achieves exponential efficiency gains while keeping spatial complexity bounded. A more detailed analysis of memory and computational overhead is provided in Appendix C.

### 6.3 Main Results on Generation

We further evaluate Fission Encoding on image generation tasks, which demand higher activation precision due to richer decoding information flow. Conventional ANN-based generators typically require at least 6-bit precision (corresponding to 64 time-steps in SNNs), making low-latency inference challenging.

We convert two pretrained models—WGAN-GP [13] and DDPM [19]—into SNNs using post-training calibration [28] and apply Fission Encoding with increasing fission rates as $T$ decreases: $[10, 30, 60, 90, 120]$ for $T$=64 to $T$=4. Directly reducing $T$ below 32 severely degrades image quality, whereas Fission Encoding recovers visually comparable results even at $T$=4, as confirmed by PSNR measurements against ANN outputs. Under low-latency settings, DDPM models outperform WGAN-GP, preserving consistency despite reduced detail. To our knowledge, no prior work reports comparable visual quality at $T \leq 8$, enabling practical generative inference on neuromorphic hardware.

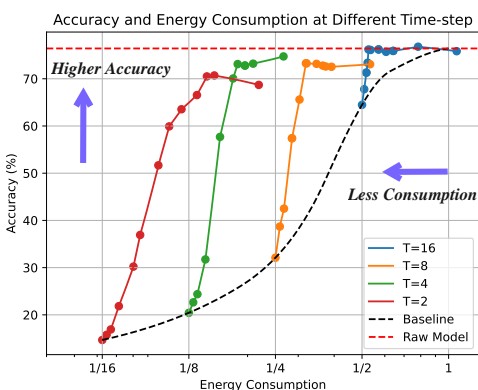

Figure 4: Accuracy and energy consumption across time-steps. Each point on the same curve denotes a different fission rate.

### 6.4 Further Analysis of Algorithm

**Fission Rate and Threshold.** Fig.4 shows that accuracy initially improves rapidly with increasing fission rate and saturates once most layers achieve sufficient precision. At very low time-steps, the saturation point shifts due to the SNN unevenness effect [17], where excessive residual potential accumulates in late steps. Layer-wise analysis in Fig.6 reveals that deeper layers typically undergo more fission ("Fission 1/2" denotes activations split into two or three neurons). Fig.7 further illustrates that high-sensitivity neurons often exhibit long-tailed or multimodal activation distributions; optimal thresholds derived after fission (red lines) effectively capture these patterns.

**Ablation and Sensitivity-Based Selection.** We further examine the effect of Stage-1 sensitivity selection within the pipeline. Unlike prior coding schemes [24, 30] that uniformly allocate precision, our approach selectively applies fission to highly sensitive neurons. As shown in Fig. 8, disabling selection (i.e., fissioning all neurons) slightly increases peak accuracy ($< 2\%$) by including rare but important units. However, this significantly increases computational and memory overhead. With selection enabled, about $70\%$ of neuron fission is avoided, yielding a better cost–accuracy trade-off.

**Comparison with Baselines.** We compare Fission Encoding with two population-coding methods: Spatial Approximation [24] and Group Neuron [30]. Even without sensitivity selection, our approach achieves $68.22\%$ accuracy with $100\%$ additional neurons, surpassing the $66.94\%$ of [24]. When each method uses its optimal neuron budget, baseline accuracy plateaus at $72.85\%$ with $400\%$ overhead, while ours reaches $73.28\%$ with only $78\%$. These results demonstrate that Fission Encoding delivers both higher accuracy and significantly better neuron efficiency.

### 6.5 Computational and Hardware Implications

**Computational Complexity.** Transitioning from cascade to parallel firing introduces more comparisons (CMPs) but significantly reduces additions (ADDs). For a group of $n$ split neurons, achieving $2^n$-level precision requires $2^n-1$ CMPs and $n$ ADDs with our approach, compared to $2^n-1$ CMPs

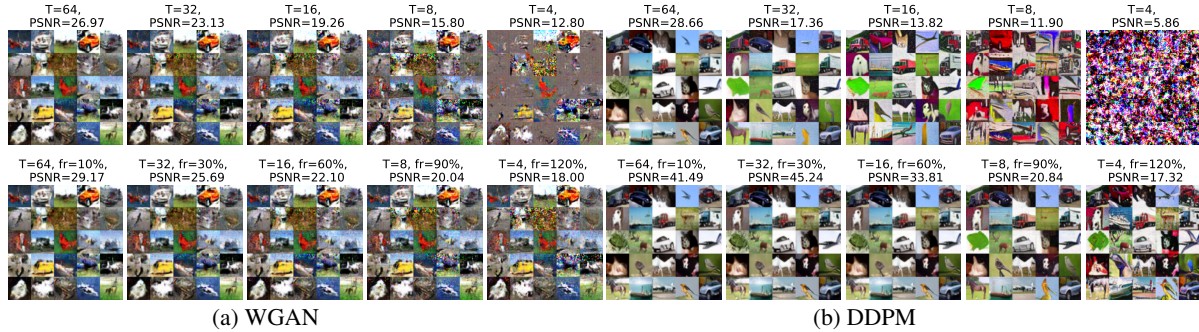

(a) WGAN            (b) DDPM

Figure 5: Image generation and PSNR for CIFAR-10 at varying inference time-steps. Top row: original SNN outputs. Bottom row: results with fission encoding applied.

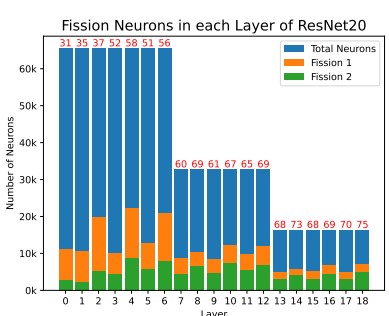

Figure 6: Neuron fission rounds across layers. Red numbers is the average fission rates per layer.

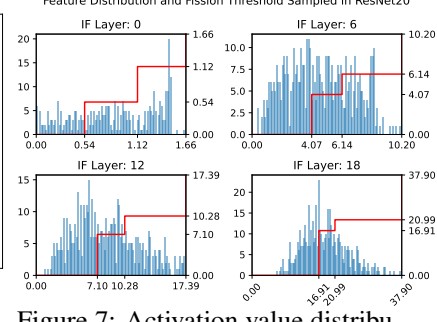

Figure 7: Activation value distributions. Red line reflects the fission thresholds and the response modes.

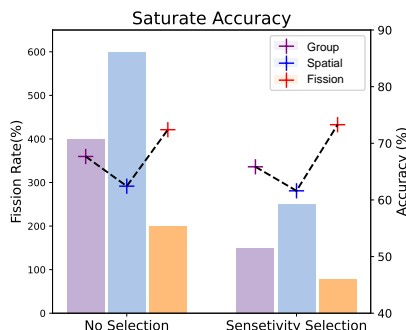

Figure 8: Comparison of accuracy and fission rate with related works and ablation of sensitivity selection.

and $2^n-1$ ADDs in conventional parallel coding. While CMP complexity grows exponentially in both cases, ADD operations—typically more expensive—scale only linearly, making computation substantially more hardware-friendly. The slightly increased CMP cost is offset by exponentially fewer time-steps, yielding overall energy and latency benefits without hardware modifications. Further complexity analysis is provided in Appendix.C.

**Compatibility with Pruning.** Fission Encoding can be combined with unstructured pruning to further reduce memory overhead by removing neurons with low sensitivity and weak post-synaptic weights. While less effective on conventional GPUs, this sparsification provides clear gains on neuromorphic hardware: on ResNet20 with $T=2$, pruning reduces memory usage from 2.25 GB to 1.53 GB with less than 2% accuracy loss.

## 7 Conclusion and Discussion

Activation quantization under limited time-steps remains a critical obstacle to scaling SNNs toward complex tasks. Drawing inspiration from post-training quantization in ANNs, we reinterpret population coding from a hardware-aware perspective and propose Adaptive Fission, a post-training mechanism that redistributes precision across the network while preserving binary spike compatibility.

Our design is motivated by a simple observation: conventional ANN quantization imposes two unnecessary constraints—global fixed bitwidth and fixed bit weights (e.g., powers of two). By representing precision through multiple spiking sub-neurons, Adaptive Fission relaxes these constraints and flexibly allocates representational capacity where it is most needed. This enables efficient conversion of ANN backbones into SNNs without additional training or fine-tuning.

While broadly applicable across architectures and tasks, our framework shares limitations with existing ANN-to-SNN conversions. It is not directly suitable for event-driven datasets (e.g., CIFAR10-DVS), where input dynamics evolve with simulation time. Moreover, the temporal gains achieved by Adaptive Fission come at the cost of increased spatial complexity, which may raise resource demands in large-scale deployments. Despite these trade-offs, we believe that this precision-aware, post-training strategy represents a practical step toward bridging the gap between high-performance ANNs and efficient neuromorphic deployment.

## Acknowledgements

This work was supported in part by the National Key Research and Development Program of China under STI 2030—Major Projects (No. 2021ZD0200300), in part by the National Key Research and Development Program of China (No. 2024YDLN0006), in part by the National Natural Science Foundation of China (Grant No. 62176133), in part by the Tsinghua University - Meituan Joint Institute for Digital Life under Agreement No. C0210322000380, in part by the Tsinghua-Fuzhou Data Technology Joint Research Institute (Project No. JIDT2024013), and in part by Qualcomm Technologies, Inc. under Statement of Work No.TSI617560.

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

# A   Detailed Proofs and Explanations

This section provides proofs and explanations for theorems and propositions in Section.3, 4 and 5.

## A.1   Explanations of Equations 5 and 6

We provide additional clarification on the motivation and interpretation of the two key equations used in ANN-to-SNN conversion preliminaries.

**Equation 5: Weight Normalization for Hardware Compatibility.**   In the weighted-spike formulation introduced in Eq. 3, the spike output $s(t)$ takes values in $\{0, \theta\}$ instead of the binary set $\{0, 1\}$ assumed in standard IF neuron models. This discrepancy causes a mismatch when deploying SNNs on neuromorphic hardware or when aligning their computation with standard ANN formulations.

A common solution in training-free ANN-to-SNN conversion [28, 42] is to absorb the threshold scaling factor into the synaptic weights. For example, when a presynaptic neuron emits a spike with magnitude $V_{th}^{(\ell-1)}$ and the postsynaptic neuron fires if its membrane potential exceeds $V_{th}^{(\ell)}$, the firing condition is

$$V_{th}^{(\ell-1)}\mathbf{W}^{(\ell)} > V_{th}^{(\ell)}. \tag{25}$$

This inequality is equivalent to

$$\frac{V_{th}^{(\ell-1)}\mathbf{W}^{(\ell)}}{V_{th}^{(\ell)}} > 1, \tag{26}$$

indicating that the threshold $V_{th}^{(\ell-1)}$ can be absorbed into the synaptic weight, thereby fixing the threshold to $1$ without changing the neuron dynamics.

Following the same principle, Eq. 5 performs a linear rescaling:

$$w := \theta w, \quad s(t) := \frac{s(t)}{\theta} = H(v(t) - \theta) \in \{0, 1\}, \tag{27}$$

which normalizes the spike outputs and ensures computational compatibility with hardware platforms that support only binary spikes. Conceptually, this step aligns the weighted-spike model with a standard $\{0, 1\}$ IF neuron model, while preserving equivalence in input-output behavior.

**Equation 6: Firing-Rate Approximation as Quantized Activation.**   Equation 6 provides an interpretation of the firing rate of a spiking neuron in terms of a quantized ANN activation. In rate-coding SNNs, the firing rate over $T$ time-steps,

$$s = \frac{1}{T}\sum_{t=1}^{T} s(t), \tag{28}$$

represents the activation intensity. Assuming that the total integrated input matches the ANN pre-activation $a$, i.e., $\sum_t q_{\text{snn}}(t) = q_{\text{ann}} = a$, the firing rate can be rewritten as:

$$s = \frac{a - v(T)}{T} = \frac{\theta \cdot \text{clip}\left(\left\lfloor \frac{a}{\theta} \right\rfloor, 0, T\right)}{T}. \tag{29}$$

This expression shows that the firing rate effectively corresponds to a $T$-level quantized version of the ReLU activation $a$. The term $\lfloor a/\theta \rfloor$ represents the discrete spike counts (bounded by $T$), while the residual membrane potential $v(T)$ reflects the quantization error. Such a formulation is widely used in quantization-based conversion approaches [28, 7], where the temporal accumulation of spikes is treated as a discretized approximation of continuous activations.

Furthermore, Eq. 6 reveals a useful design principle: since the quantization granularity is determined by $\theta$, a larger simulation window $T$ allows for a smaller threshold $\theta \approx \max(a)/T$, thereby enabling finer-grained representation of neuronal activation with minimal quantization error.

## A.2 Additional Proof for Population Encoding Models

*Proof for Prop.4.2.* This is a special case of Prop.4.3 when the bit-length of a neuron group $k = 1$. See the proof of Prop.4.3 $\qquad\square$

*Proof for Prop.4.3.*

Define $\left\{ s_{(1)}, \ldots, s_{(m)} \middle| s_{(i)} \in \mathcal{S}, s_{(1)} \leq \cdots \leq s_{(m)} \right\}$, where $m = (T+1)^k$ as an ordered sequence of all possible spike combinations. Consider $q_{(i)} \in \left[ s_{(i-1)}, s_{(i+1)} \right)$. The errors within this interval can be calculated as:

$$
\begin{aligned}
r_{(i)} &= \int_{s_{(i-1)}}^{s_{(i)}} (q - s_{(i-1)}) dq + \int_{s_{(i)}}^{s_{(i+1)}} (q - s_{(i)}) dq \\
&= \left[ s_{(i)} - \frac{1}{2} (s_{(i-1)} + s_{(i+1)}) \right]^2 + \frac{1}{4} \left[ s_{(i-1)} - s_{(i+1)} \right]^2 \\
&\geq \frac{1}{4} \left[ s_{(i+1)} - s_{(i-1)} \right]^2,
\end{aligned}
$$

where the equation is taken when $s_{(i)} = \frac{1}{2}(s_{(i-1)} + s_{(i+1)})$. Define $s_{(0)} = 0, s_{(m+1)} = q_{\max}$ as boundary conditions,

$$
\begin{aligned}
r &= \frac{1}{2q_{\max}} \left( \sum_{i=1}^{m} r_{(i)} + \frac{1}{2}(s_{(1)} - s_{(0)})^2 + \frac{1}{2}(s_{(m+1)} - s_{(m)})^2 \right) \\
&\geq \frac{1}{8q_{\max}} [2(s_{(1)} - s_{(0)})^2 + (s_{(2)} - s_{(0)})^2 + \cdots + (s_{(m+1)} - s_{(m-1)})^2 + 2(s_{(m+1)} - s_{(m)})^2] \\
&= \frac{1}{8q_{\max}} \cdot \frac{4(s_{(m+1)} - s_{(0)})^2}{(\frac{1}{2} + 1 + \cdot + \frac{1}{2})} \\
&= \frac{1}{8q_{\max}} \cdot \frac{4q_{\max}^2}{(T+1)^k} \\
&= \frac{q_{\max}}{2(T+1)^k}
\end{aligned}
$$

where the equation is taken when $s_{(i)} = \frac{1}{2}(s_{(i-1)} + s_{(i+1)})$ holds for all $i = 1, \ldots, m$. Thus, we have

$$
b = s_1 = \frac{q_{\max}}{2(T+1)^k},
$$
$$
\theta_i = \frac{q_{\max}}{(T+1)^i}
$$

$\qquad\square$

*Proof for Prop.4.4.* Consider multiple neurons operating at the final single step, setting $T := 1$ and $q := v_{temp}(T)$. Substitute these into Eq.9 to obtain the above result. $\qquad\square$

## A.3 Proof for Theorem.5.1

*Proof.* For clarity, we let $\theta_{old} = 1$ and denote $\theta_{k+1} = \theta$, consider $v \in [0, \theta_k')$ in most cases. The reduction caused by fission is given by:

$$
\Delta v = v - v' = \begin{cases} 0 & 0 \leq v < \theta \\ \theta & \theta \leq v < 1 - \theta \\ 1 - \theta & 1 - \theta \leq v < 1 \end{cases}
$$

Using the CDF to represent the piece-wise function, we can rewrite the error reduction on sample batch. The optimization target is defined as

$$\mathbb{E}[\Delta v] = \int \Delta v \cdot p(v)dv$$

$$=\theta \int I_{\theta \leq v < 1-\theta} \cdot p(v)dv - (1-\theta)\int I_{1-\theta \leq v < 1} \cdot p(v)dv$$

$$=\theta \int_{\theta}^{1-\theta} p(v)dv - (1-\theta)\int_{1-\theta}^{1} p(v)dv$$

$$= [F(1-\theta) - F(\theta)]\,\theta + [1 - F(1-\theta)]\,(1-\theta).$$

Given $\theta = \arg\max_\theta \mathbb{E}[\Delta v]$, the extremum can be obtained by derivation. At the maximum point, we have:

$$\frac{d\mathbb{E}[\Delta v]}{d\theta} = [2F(1-\theta) - F(\theta) - 1] - \theta \cdot [2p(1-\theta) + p(\theta)] + p(1-\theta) = 0,$$

where $p$ is the probability density function. Considering that in actual sampling, the distribution of $v$ is approximately discrete, we assume $p(\theta) = p(1-\theta) = 0$. Thus we have:

$$2F(1-\theta) - F(\theta) - 1 = 0$$
$$\mathbb{E}\Delta v = 1 - F(1-\theta) = F(1-\theta) - F(\theta)$$

yielding the result in Eq.18. We can further estimate $\mathbb{E}v$ by piece-wise scaling

$$0 \leq \int_0^\theta v dv \leq \theta F(\theta),$$

$$\theta\,[F(1-\theta) - F(\theta)] \leq \int_\theta^{1-\theta} v dv \leq (1-\theta)[F(1-\theta) - F(\theta)],$$

$$(1-\theta)(1 - F(1-\theta)) \leq \int_{1-\theta}^1 v dv \leq 1 - F(1-\theta).$$

Substituting this to $\mathbb{E}[v]$ and $\mathbb{E}[\Delta v] = 1 - F(1-\theta)$ back to $\mathbb{E}v$ can derive the range for $\mathbb{E}[\Delta v]$.  □

---

**Algorithm 1** Overall Algorithm

---

**Input**: Original SNN; time-step $T$

   Set fission rate $f$, required accuracy $p\%$.
   Collect a batch of input data $x^{(j)}$
   **while** Validation accuracy $< p\%$ **do**
      **for** each feature dimension $fea$ **do**
         Conduct SNN forward propagation for $T$ steps to calculate residual potential $v_{fea}(T)$.
         Conduct ANN forward and backward propagation to calculate gradient $\frac{da_{out}}{da_{fea}}$.
      **end for**
      Calculate sensitivity $\boldsymbol{\sigma}_l$ with $v_{fea}(T)$ and $\frac{da_{out}}{da_{fea}}$. (cf. Eq.16)
      Determine threshold $\sigma_{th}$ as the $f$-percentile of all $\boldsymbol{\sigma}_l$
      **for** each layer $l = 1, 2, \ldots, m$ in the SNN **do**
         **if** sensitivity of $i$-th neuron $\sigma_{li} \geq \sigma_{th}$ **then**
            Perform fission encoding (cf. Theorem.5.1) and obtain new threshold sets $\boldsymbol{\theta}_{li}$ of post-fission neuron group
         **end if**
      **end for**
   **end while**
**Output**: Threshold sets $\boldsymbol{\theta}_{li}$ for all fission neurons

---

## B   Guidelines for the Overall Pipeline

We first provide the algorithm for our pipeline described in Section.5. The fission rate $f$ should be dynamically adjusted based on the network architecture and the desired reduction in time-steps. For modest reductions in time-steps (e.g., halving the time-steps), setting $f \approx 15\%$ is typically effective. For more substantial reductions, $f$ can be incrementally increased by 15%, and further tuning may be required based on empirical performance. In the case of an ANN-to-SNN conversion with 32 time-steps, where the inference requires at least 8 time-steps (i.e., reducing to one-quarter of the original time-steps), a higher threshold of $f \approx 80\%$ is usually sufficient. This can often be achieved with just one round of fission encoding. However, for more aggressive reductions, where fewer time-steps are required (e.g., reducing to 2 or 4 steps), the fission rate $f$ should be lowered to around 35%, and multiple rounds (typically 2 to 4 iterations) of fission encoding may be necessary to ensure performance convergence. The fission rate should be continuously adjusted based on the specific network architecture and the corresponding trade-offs between computational efficiency and accuracy. Empirical evaluation is recommended for fine-tuning these parameters in different settings.

## C   Spatial Complexity Analysis

We analyze the spatial and computational implications of neuron fission from four complementary perspectives:

**Memory Overhead.** Consider a fully connected layer $Linear(n, n)$ with $n$ thresholds, $n$ membrane potentials, and $n^2$ synaptic weights. After fission with an average rate $k$, each original neuron is split into $k + 1$ neurons. These fissioned neurons *share the same input* (membrane potential) but have *independent outputs*, which determines how overhead arises:

- *Thresholds:* Increase from $n$ to $n(k + 1)$, one per fissioned neuron.
- *Membrane Potentials:* Theoretically remain $n$ since inputs are shared, but in practice duplicated to $n(k + 1)$ for matrix computation efficiency.
- *Synapses:* Each new neuron contributes $n$ additional outgoing synapses, resulting in an increase of $nk \times n$ weights.

Thus, memory overhead grows **linearly** with the fission rate $k$ across all components. In deployment, compiler optimizations such as in-place computation and buffer reuse typically reduce actual memory usage below this theoretical bound.

**Computation Complexity (CMPs and ADDs).** The transition from cascade to parallel firing indicators introduces additional comparisons but dramatically reduces additions. For a group of $n$ split neurons, the parallel model requires $2^{i-1}$ comparisons (CMPs) and one addition (ADD) for the $i$-th neuron, leading to a total of $2^n - 1$ CMPs and $n$ ADDs for $2^n$-level precision. In contrast, achieving the same precision with a conventional parallel scheme would require $2^n - 1$ neurons, each with one CMP and ADD, resulting in $2^n - 1$ CMPs and $2^n - 1$ ADDs overall. While CMPs grow exponentially in both cases, ADDs — typically more expensive — grow only linearly in our method. Although a cascade implementation would reduce comparisons to $n$, its sequential nature hinders parallel execution and is thus unsuitable for hardware deployment.

**Hardware Implications.** Threshold comparisons align well with neuromorphic architectures, where CMP circuits are often implemented as dedicated units due to inherent spiking dynamics. Comparators typically consume 3–5$\times$ less energy and have lower delay and area than adders. As a result, the CMP-dominant computation pattern introduced by Adaptive Fission is well matched to hardware execution. Moreover, per-step complexity increases are offset by exponential reductions in time-steps, yielding overall computational efficiency gains.

**Power and On-Chip Efficiency.** Measured power profiles show that a significant portion of neuromorphic ASIC energy is consumed by I/O and memory access rather than arithmetic computation. Consequently, the growth in CMP operations has limited impact on total power consumption, while reduced time-steps translate into substantial energy savings. This trade-off — slightly higher per-step complexity for significantly improved temporal efficiency — is generally advantageous for neuromorphic deployment.

**Hardware Compatibility.** All neurons, including fissioned ones, are implemented as modular IF units with segmented step functions expressed via CMP-based firing logic. Spikes remain binary

$\{0, \theta_i\}$ and are normalized to $\{0, 1\}$ during deployment. Hardware experiments on our prototype neuromorphic chip confirm that Adaptive Fission integrates seamlessly without requiring custom logic paths.

# D    Discussion on spike-based ViTs and LLMs

Our method is applicable to spike-based vision transformers, as demonstrated with the Spike-driven Transformer in the main paper. However, many existing models still include floating-point operations (e.g., Softmax), limiting full compatibility with neuromorphic hardware. We believe that future spike-based transformers should avoid such nonlinearities to enable efficient deployment. We further tested our method on the Spatio-temporal Approximation model [24], a floating-point-free ViT-B/32 backbone from CLIP, used for zero-shot classification. Results on both GPU and the HP201 ASIC are shown in Table 2. While basic operations executed successfully, limited compiler support led to low execution efficiency, preventing meaningful power measurements.

| Method | Time-Step | Fission Rate | Accu. (%) |
|---|---|---|---|
| CIFAR-100 & CLIP-pretrained ViT-B/32 Zero-shot | | | |
| STA | 64 | No | 85.25 |
| | 32 | | 84.15 |
| **+ Fission** | 16 | 0.53 | 82.90 |
| | 8 | 1.15 | 81.48 |

Table 2: Accuracy comparison of pretrained ViT-B/32 by CLIP for Zero-shot classification.

Applying our method to large language models (LLMs) faces additional challenges:

**Attention precision:** Attention scores typically require at least 8-bit precision, translating to long inference durations in SNNs. Adaptive Fission can help alleviate this, though memory costs may become more noticeable.

**Error accumulation:** The scale of LLMs amplifies small precision errors. More refined fission rate allocation strategies may be needed across network depth to balance overhead and accuracy.

# E    Experiment Settings

### E.1    Hardware Resources

All our experiments were conducted on a workstation with Intel-13900KS, 32GB memory with a single 4090 GPU and a Lynxi HP201 neuromorphic accelerator. As a commercial version of Tianjic chip [34], HP201 adopts a many-core decentralized architecture, commonly utilized in neuromorphic chips, and integrates 60 configurable functional cores, 16 GB of memory, and image encoding/decoding units. This design allows it to support quantized ANN/SNN models of up to approximately 13 billion parameters. The chip is packaged in a PCIe form factor with a peak power consumption of 55 W. The provided LynBIDL library enables the compilation of SNN models based on PyTorch. However, since the underlying interfaces are not open, we are unable to directly measure the power consumption of individual cores or the number of MAC/AC operations during runtime. Instead, we record the total energy consumption using power readings from the sensors. While this method inevitably introduces interference (including power consumption from peripheral circuits, memory, and I/O interfaces), it reflects a more practical scenario. In such a case, reducing inference time significantly contributes to lowering power consumption, as the energy usage of peripheral circuits is largely determined by runtime duration.

Notably, while fission encoding can increase the number of neurons for single activation value, it does not alter the network's original parameters. As a result, the memory consumption typically increases by less than twice that of the baseline model.

### E.2 Implementation of Classification

The baseline models and network architectures used in fission are directly sourced from the original codebase, including the Calibration [28] and QCFS [4] conversion methods, and spike-driven transformers [49]. We slightly modify the TEBN [10] hyperparameters to suit ResNet20 and VGG16, as the original experiments were conducted on ResNet19 and VGG11. Details can be found in the code.

### E.3 Implementation of Image Generation

For the image generation model, we employ a 4-layer convolutional architecture based on the improved Wasserstein GAN (WGAN-GP) [13]. The architecture is structured as follows:

- ConvTranspose2d(100, 1024), BatchNorm2d, ReLU,
- ConvTranspose2d(1024, 512), BatchNorm2d, ReLU,
- ConvTranspose2d(512, 256), BatchNorm2d, ReLU,
- ConvTranspose2d(256, 3)

All convolutional transpose layers use a kernel size of 4. For the first layer, the stride is set to 1 and padding to 0, while for the remaining layers, the stride is 2 and padding is 1. This model is trained on the CIFAR-10 dataset using the WGAN-GP framework [13]. After training, the model is converted to a spiking neural network (SNN) representation by discretizing the output into 64 time-steps using the Calibration (Calib.) method.

It is important to note that direct training of SNNs for image generation tasks, especially with lower quantization precision, typically fails to achieve satisfactory performance, as demonstrated by the difficulty in generating high-quality images. Consequently, methods such as fission encoding are employed to improve inference performance in this context.

For the diffusion model, we follow the implementation described by Ho et al. [19]. The model consists of 4 residual blocks with a channel configuration of [2, 4, 4, 2], and uses the Swish activation function instead of ReLU. This prevents a direct conversion of the model to an Integrate-and-Fire (IF) spiking neuron network. To address this, we introduce an IF layer after each Swish activation, thereby discretizing the continuous output into spike events.

The diffusion model is trained on CIFAR-10 with 500 iterations for image generation. After training, the model is also converted to 64 time-steps using Calib. However, it is worth noting that the ANN-based diffusion model requires 500 iterations for each image generation, which results in a corresponding SNN model requiring $64 \times 500 = 32,000$ iterations. This computational cost is prohibitively high for real-time inference and necessitates the use of fission encoding to reduce latency.

For evaluating experimental results, we used the Peak Signal-to-Noise Ratio (PSNR) to measure the difference between the generated images and those from the original ANN model. The PSNR is calculated as follows:

$$PSNR = 10 \cdot \log_{10} \left( \frac{MAX^2}{MSE} \right), \tag{30}$$

where $MAX$ represents the maximum possible pixel value in the image data. For 8-bit images, $MAX = 255$. The Mean Squared Error (MSE) is defined as the average of the squared differences between the pixel values of the generated image and the reference image, and is calculated as follows:

$$MSE = \frac{1}{mn} \sum_{i=1}^{m} \sum_{j=1}^{n} [I(i,j) - K(i,j)]^2, \tag{31}$$

where $I(i,j)$ and $K(i,j)$ represent the pixel values at position $(i,j)$ in the generated image and the reference image, respectively, and $m$ and $n$ denote the height and width of the images.

### E.4 Random error and Reproducibility

Fission Encoding employs a small batch of examples for Sensitivity detection and estimates each neuron's cumulative density function, which introduces some variability. By allocating an additional validation set, we can assess the optimal execution, which is then applied to the test set. In most of our experiments, we selected the best solution from 10 runs on the validation set. Table 3 displays the best and worst cases under these conditions, highlighting relatively significant performance variability.

| Time-step | 10% | 20% | 40% | 80% | 160% | 320% |
|---|---|---|---|---|---|---|
| T=2 | 12.14-15.77 | 12.02-16.91 | 17.76-21.84 | 25.14-30.22 | 48.79-51.67 | 60.43-66.55 |
| T=4 | 14.06-22.65 | 18.10-24.38 | 27.69-31.74 | 50.67-57.69 | 67.35-72.80 | 70.44-74.73 |
| T=8 | 35.51-38.70 | 36.32-42.53 | 52.69-57.40 | 68.11-73.26 | 69.32-72.54 | 70.68-73.07 |
| T=16 | 68.65-71.43 | 72.74-76.10 | 73.31-76.21 | 73.55-75.90 | 72.42-76.79 | 74.51-75.84 |

Table 3: The best and worst accuracy range of fission rate at different time-steps within 10 runs.

## F  Raw Data for Figures

In this section, we present the raw data corresponding to the figures in the main paper. The data points in Fig.4 are provided by Table 1 along with the supplementary Table 4 here. Table 5 corresponds to Fig.8.

| Time-step | 10% | 20% | 40% | 80% | 160% | 320% |
|---|---|---|---|---|---|---|
| T=2 | 15.77 | 16.91 | 21.84 | 30.22 | 51.67 | 66.55 |
| T=4 | 22.65 | 24.38 | 31.74 | 57.69 | 72.80 | 74.73 |
| T=8 | 38.70 | 42.53 | 57.40 | 73.26 | 72.54 | 73.07 |
| T=16 | 71.43 | 76.10 | 76.21 | 75.90 | 76.79 | 75.84 |

Table 4: Accuracy comparison of fission rate at different time-steps. The raw model is ResNet20 trained with 32 time-steps on CIFAR-100 with calibration.

| Method | Fission (%) | Accuracy |
|---|---|---|
| Group [30] | 100 | 66.94 |
| | 400 | 71.67 |
| Sensitivity + Group | 100 | 69.39 |
| | 150 | 72.85 |
| Spatial [24] | 100 | 52.51 |
| | 600 | 66.43 |
| Sensitivity + Spatial | 100 | 63.37 |
| | 250 | 68.61 |
| Fission (Ours) | 100 | **68.22** |
| | **200** | 72.40 |
| Sensitivity + Fission | 100 | **72.79** |
| | **78** | 73.28 |

Table 5: Ablation and Comparison of different neuron selection and group combining methods.

## G  Experimental Visualization and Analysis

This section provides a detailed analysis of the various components of the fission encoding method, including demonstrations of its effects on neurons within actual networks, the distribution of neurons across different layers after fission, and the distribution of sensitivity. Additionally, we provide more results of image generation tasks in a high-resolution format.

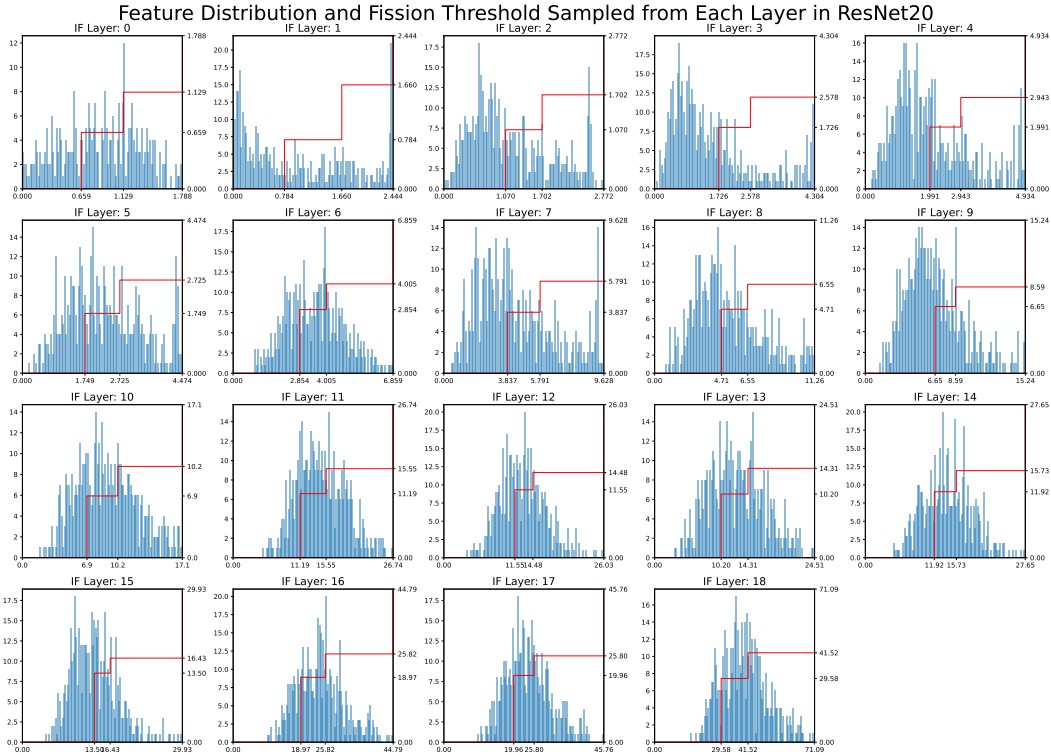

Figure 9: Feature distribution and their fission threshold in ResNet20. The feature distributions presented are randomly sampled from the more sensitive neurons across various layers of the network. The horizontal axis represents the feature distribution, the left vertical axis (associated with the bar chart) indicates the distribution density, while the right vertical axis (corresponding to the red line) reflects the fission threshold and the activation function resulting from the combination of neurons post-fission.

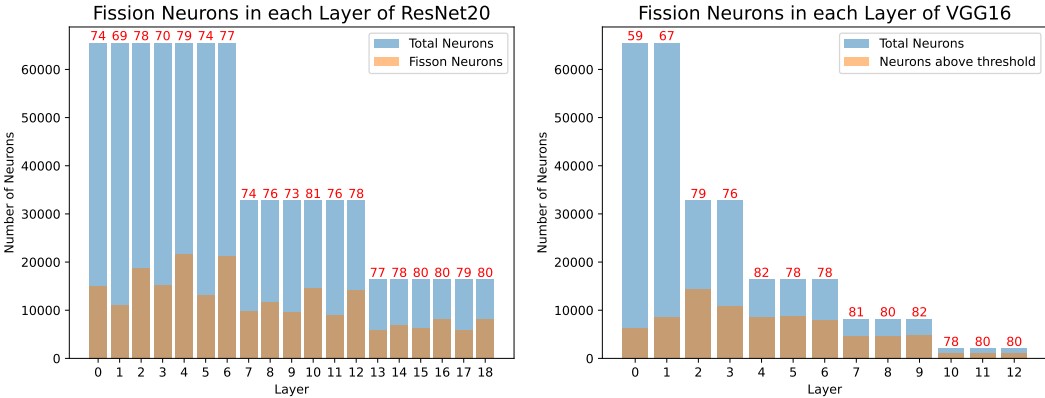

Figure 10: Distribution of Fission Neurons on all ReLU activation layers in ResNet20 and VGG16, when 30% of neurons undergo fission in a single round. The distribution shows that although the number of activation values significantly decreases in the network, the number of fission neurons only slightly reduces, while their proportion increases in deeper layers.

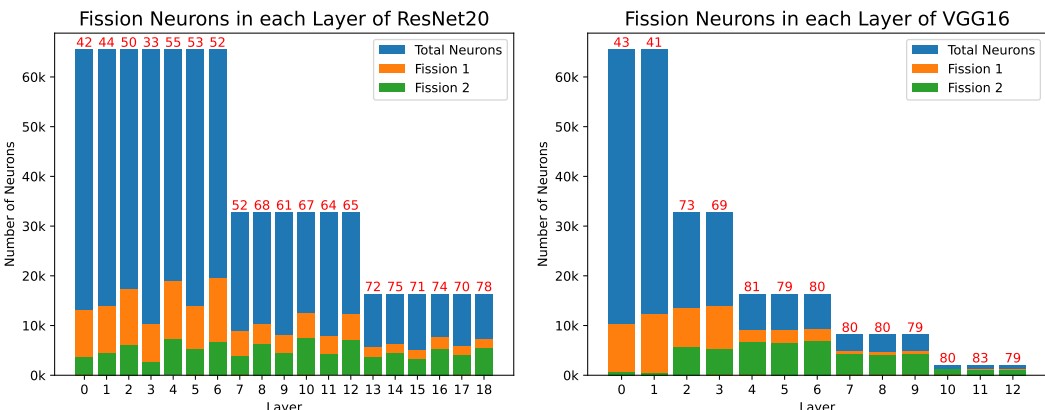

Figure 11: Distribution of Fission Neurons across ReLU activation layers in ResNet20 and VGG16 after two rounds of fission. The data indicates that the proportion of neurons undergoing multiple fission rounds is higher in the deeper layers, suggesting an increasing demand for feature precision in the deeper network layers.

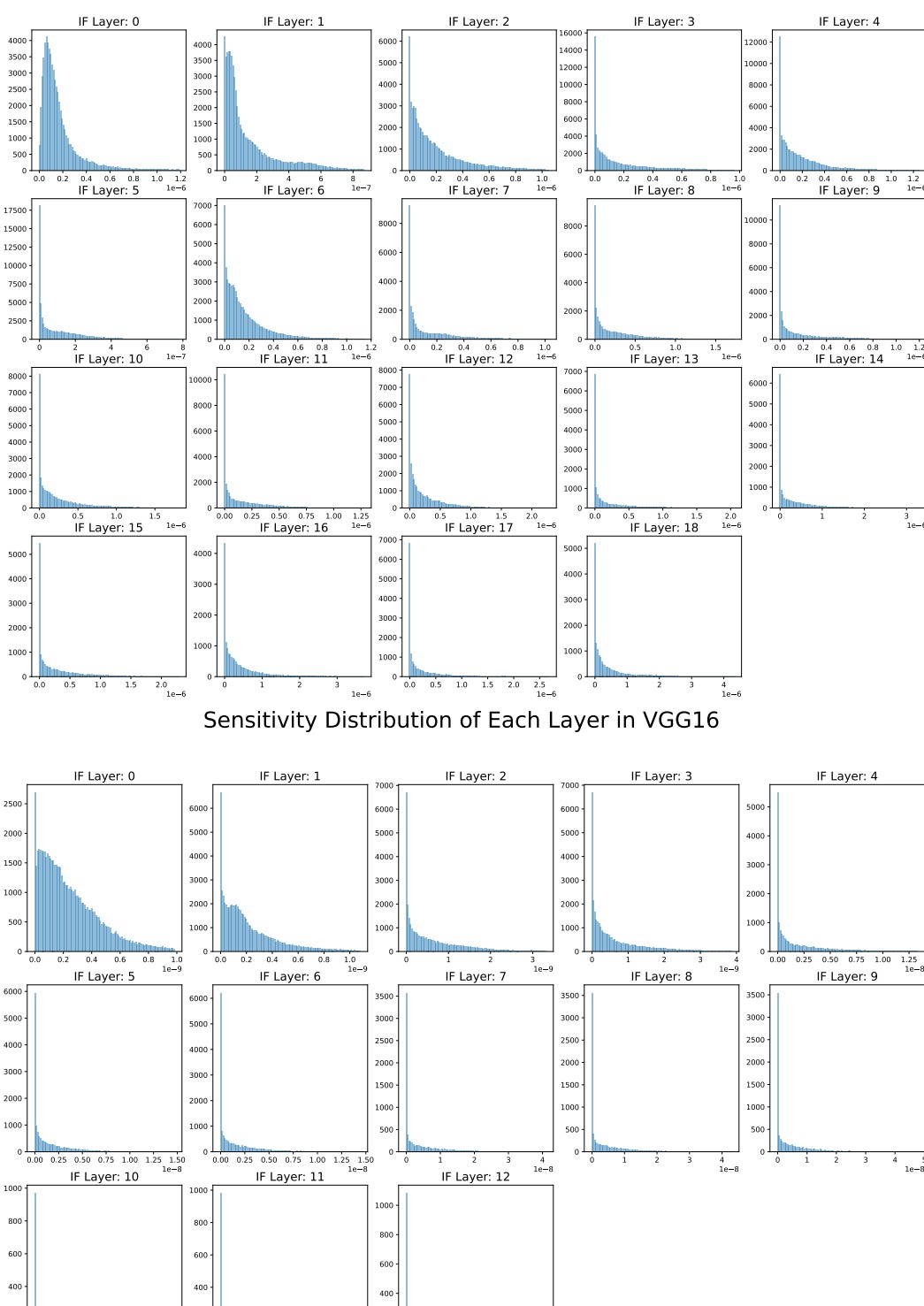

Figure 12: Sensitivity in ResNet20 and VGG16. The distributions is characterized by a long-tail distribution. The majority of neurons have a sensitivity of zero, indicating that fission is unnecessary for them. However, due to variations in sample sampling, there may be some random error in this distribution.

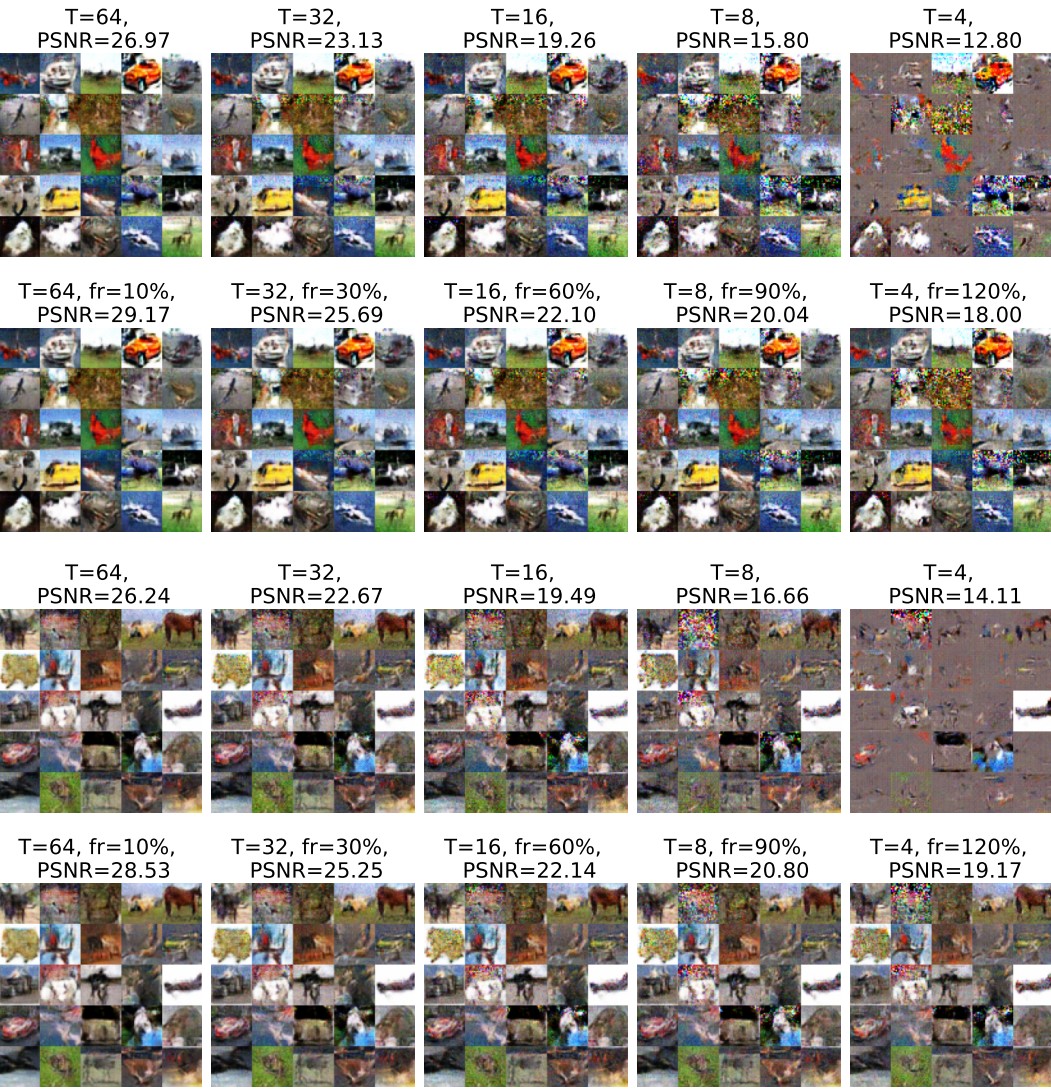

Figure 13: Image generation results of WGAN and PSNR for CIFAR-10 at different inference time-steps.

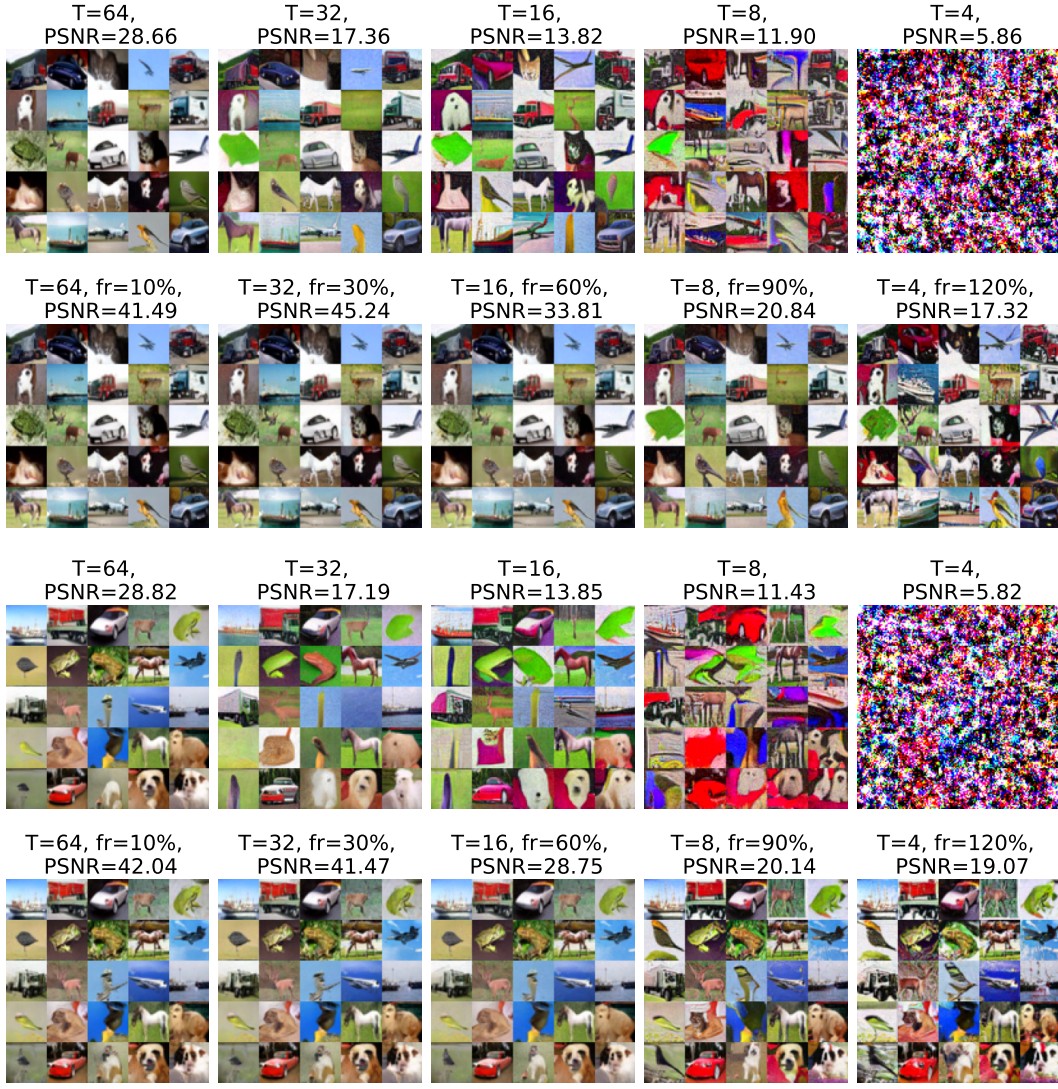

Figure 14: Image generation results of diffusion U-Net and PSNR for CIFAR-10 at different inference time-steps.

