# OpenReview forum: "Adaptive Fission: Post-training Encoding for Low-latency Spike Neural Networks"
_NeurIPS.cc/2025/Conference — NeurIPS 2025 poster_

### Official Review · Reviewer_QkMA · 2025-06-25

**Clarity:** 3
**Significance:** 4
**Originality:** 4
**Rating:** 5
**Confidence:** 4

**Summary:**

This paper proposes training of a dynamic threshold for population encoding SNN model. By allocating a larger number of neurons to error-sensitive activation, they can reduce timesteps with minimized computation overhead. Demonstrating on a neuromorphic chip, this work reduces energy consumption on VGG16, ResNet20, and Transformer.

**Questions:**

1.	Within the implemented hardware, does it merge all $s_i(t)$ into $s(t)$ as in (14) and then pass the spikes to the next layer? Or, are all $s_i(t)$ in the group passed to the next layer in binary (spike) format, and instead the weights pre-multiplied by the thresholds $\theta_i(t)$ are exploited on the synaptic operation of the next layer?

2.	Can you show how the increased memory overhead (threshold, potential, synaptic storage) is composed by the fission rate? I believe that such a quantitative analysis would further emphasize the paper’s strength.

3.	I recommend validating performance with ResNet-based models for the ImageNet benchmarks in Table 1. Presenting performance on ImageNet only with the very large VGG can diminish the contribution of this paper.

**Ethical Concerns:**

["NO or VERY MINOR ethics concerns only"]

**Final Justification:**

This paper proposes a novel method for grouping neurons that increases performance with minimized hardware overhead. This paper demonstrates the effectiveness of its approach by showing high accuracy in various benchmarks while demonstrating that the overhead incurred by actual neuromorphic hardware is not significant. Therefore, I believe that this paper presents strategies that various SNN models requiring accuracy improvement should consider.

**Limitations:**

yes

**Paper Formatting Concerns:**

No concerns

**Quality:**

4

**Strengths And Weaknesses:**

Strength

1.	This paper implements a dynamic number of neurons per activation, considering the saliency of activations.

2.	This paper presents power consumption using the Lynxi HP201 chip.

3.	This paper only needs calibration, not post-training. And their proposed training method is reasonable and can be adapted to various SNN works.

Weakness

1.	As this work uses multiple thresholds (neurons) per activation, the overhead from potential and output synapses becomes large.

2.	The analysis with comparison baselines is insufficient. For example, group-neuron based SNN works [32] should be compared in terms of accuracy and energy. Furthermore, a comparison with A2SNN, which has reduced timesteps [1*] through post-training, may better illustrate the strength of this paper.

[1*] N. Rathi and K. Roy, “DIET-SNN: A low-latency spiking neural network with direct input encoding and leakage and threshold optimization,” IEEE Trans. Neural Netw. Learn. Syst., vol. 34, no. 6, pp. 3174–3182, Jun. 2023.

---

> ### Author Rebuttal · Authors · 2025-07-28
>
> We sincerely thank the reviewer for the detailed and constructive feedback. We hope our responses address your concerns regarding the validation of the proposed algorithm. We will include additional experimental results in the appendix if given the opportunity.
>
> ---
>
> ### **Weakness 1: Overhead of Additional Neurons**
>
> We acknowledge that increasing the number of neurons per activation may introduce additional overhead. Since post-training techniques are often considered a "no free lunch" solution, our method is to **minimize** the overhead while achieving substantial latency and energy improvements. It only selectively and sparsely introduces additional neurons to a small subset of high-sensitivity neurons, avoiding the redundancy caused by uniformly allocating bit-widths to all activations and synapses in conventional quantization.
>
> ---
>
> ### **Weakness 2.1: Comparison with Group Neuron Baseline**
>
> We provide comparative results with [32] in Appendix Table 5. For clarity, we summarize two key cases:
>
> - **Without sensitivity selection**, using the same 100% additional neurons, our method achieves **68.22%** accuracy, compared to **66.94%** for [32].
> - **With sensitivity selection**, allowing each method to use its own optimal number of neurons, [32] saturates at **72.85% accuracy** with extremely **400%** additional neurons, whereas our method achieves higher **73.28%** using only **78%** neurons.
>
> These results demonstrate that our method is both more efficient and more accurate in leveraging population-based encoding.
>
> ---
>
> ### **Weakness 2.2: Comparison with Post-Training Methods**
>
> We appreciate the suggestion and have initiated experiments to compare with DIET-SNN [1*]. Admittedly, there exists a fundamental gap of accuracy between our training-free approach and post-training methods like [1*]. The comparison experiments are still ongoing, and we provide some preliminary results.
>
> In their original paper, the ANN accuracy on VGG16-ImageNet is reported as 70.08%, and the optimized SNN accuracy after post-training reaches 69.00% at $T = 5$.
> Since their provided conversion strategy is not optimal for training-free, we adopt our own threshold selection method, and achieves an initial SNN accuracy of only **32.43%** at $T = 5$ — indicating a severe accuracy drop of training-free methods from the ANN baseline.
> Despite this significant degradation, we were able to recover the accuracy to **62.84%** in our preliminary attempts, with **2.19×** additional fission neurons. Further experiments are still in progress.
>
> ---
>
> ### **Question 1: Spike Propagation and Equation (14)**
>
> Your understanding is accurate — the hardware implementation follows exactly the **second approach** you described, i.e., all $s_i(t)$ are binary and passed through individual synapses.
>
> Equation (14) is just a conceptual illustration, but since $s(t)$ is not binary, it cannot directly participate in spike-based synaptic operations. Actually, all split neurons in the fission group emit their binary spikes individually and independently to the next layer.
> In practice, we **duplicate** the synaptic weights associated with split neurons and **scale** them by their respective thresholds $\theta_i$. These updated weights are compiled into an expanded matrix before deployment. This compilation step accounts for the main increase in memory overhead introduced by Adaptive Fission.
>
> ---
>
> ### **Question 2: Memory Overhead and Fission Rate**
>
> We provide some brief conclusions in Appendix C, and a more detailed explanation here.
>
> Consider a $Linear(n, n)$ fully connected layer. The original layer contains:
> - $n$ thresholds,
> - $n$ membrane potentials,
> - $n^2$ synaptic weights.
>
> Assume the average fission rate $k$ is evenly distributed, i.e., each neuron is split into $k+1$ neurons. It is important to note that the fissioned neurons **share the same input** (membrane potential) but have **independent outputs**, which affects how overhead arises:
>
> - **Thresholds**: Increase from $ n $ to $ n(k+1) $, one for each fission neuron.
> - **Membrane Potentials**: Though theoretically $ n $ (since all split neurons share the same potential), we allocate $ n(k+1) $ potentials in practice with duplicated channels for shared values to achieve efficient matrix operations.
> - **Synapses**: The $nk$ newly created neurons retain the same inputs (since inputs are not split) but produce distinct outputs. Therefore, each of the $nk$ neurons still contributes $n$ additional synapses to the next layer, resulting in an increase of $nk \times n$ synaptic weights.
>
> Thus, the overhead scales **linearly** with the fission rate $ k $ across all components.
>
> In real-world deployment, the actual memory growth is typically **less than the theoretical bound** due to hardware/compiler optimizations such as in-place computation and reuse of intermediate buffers. Further analysis on memory occupation component may involve an in-depth study on the compilation process.
>
> ---
>
> ### **Question 3: Evaluation for ResNet on ImageNet**
>
> We have added **ResNet-34 experiments** on ImageNet using both Calibration and QCFS. Preliminary results show comparable or better improvements than VGG16, with similar fission rates.
>
> | Method             | Time Step | Fission Rate | Accu. (%) | Mem. (GB) | Time/Epoch(s) | Energy (Wh) |
> |--------------------|-----------|--------------|-----------|-----------|----------------|-------------|
> | **ImageNet & ResNet34**                                                                              |
> | Calib. (Conversion)| 32        | No           | 64.82   | 2.68      | 348            | 3.02        |
> |                    | 16        |              | 56.21     |           | 189            | 1.73        |
> |                    | 8         |              | 31.75     |           | 110            | 0.95        |
> | **+ Fission**      | 16        | 0.17         | **66.94**     |2.81      | 214            | 2.01        |
> |                    | 8         | 0.96         | 63.75     | 4.60      | 135            | 1.38        |
> |                    | 4         | 1.72         | 60.43     | 5.57     | **55.2**         | **0.61**    |
> | QCFS. (Quantization)| 32       | No           | **69.37**   | 2.48     | 372           | 3.29       |
> |                    | 16        |              | 59.35     |           | 228            | 2.08        |
> | **+ Fission**       | 16        | 0.28         | 67.93     | 2.96     | 240            | 2.41        |
> |                    | 8         | 1.04         | 66.51     | 4.38      | 156            | 1.54      |
> |                    | 4         | 1.95         | 62.41     | 5.05      | **81.3**         | **0.95**    |
>
> Due to time constraints, directly trained SNNs are still under evaluation. Despite this, sensitivity maps and per-layer fission distributions (Appendix Figures 10 & 11) show consistent trends across VGG and ResNet, suggesting strong generalizability.
>
> As an interesting note, although VGG16 appears larger, the convolutional backbone has **fewer parameters** than ResNet-34 (14.7M vs. 21.3M). Most of VGG’s size stems from its final fully connected layers (~123M parameters).

---

> > ### Comment · Reviewer_QkMA · 2025-08-04
> >
> > Thank you for your detailed response. My concerns are now resolved, and I believe that the method proposed in this paper is novel and can be applied to various SNNs. A notable strength of this paper is that it eliminates concerns about computational overhead by implementing it on actual hardware. This paper is good enough to be accepted so that I will raise the score.

---

> > > ### Author Response · Authors · 2025-08-04
> > > **Thanks for your support**
> > >
> > > We sincerely thank the reviewer for the positive feedback and for recognizing the novelty and practicality of our method. Your support is greatly appreciated.

---

### Official Review · Reviewer_W46Z · 2025-07-02

**Clarity:** 3
**Significance:** 3
**Originality:** 3
**Rating:** 4
**Confidence:** 4

**Summary:**

This paper proposes an ANN2SNN method, call ​adaptive fission​​, a post-training encoding technique for SNNs that ​dynamically allocates precision​​ by selectively splitting sensitive neurons into groups with different scales/weights, significantly reducing latency and energy consumption (up to 80%) without performance loss on neuromorphic hardware.

**Questions:**

**Question 1: Computational Cost of Parallel Model Indicators**
What is the computational overhead associated with generating the binary indicators $\mathbb{I}_i$ in the Parallel Model implementation of neuron groups? Specifically, does the exponential growth in comparisons ($2^{i-1}$ per neuron $i$)?

**Question 2: Impact of Sensitivity-Based Selection**
If Stage 1 is not used to select sensitive neurons—and threshold fission is instead applied to all neurons—what would be the resulting accuracy implications and computational cost?

**Question 3: Comparison with State-of-the-Art Methods**
The proposed fission technique is not compared with state-of-the-art works, such as:
CNN Methods:
- *Reducing ANN-SNN conversion error through residual membrane potential*  2023
Transformer Methods:
- *SpikeZIP-TF: Conversion is all you need for transformer-based SNN*  2024
- *Towards high-performance spiking transformers from ANN to SNN conversion* 2024
- *Spatio-temporal approximation: A training-free SNN conversion for transformers* 2024
I wonder what will be the result if the fission method is applied to the sota work?

**Ethical Concerns:**

["NO or VERY MINOR ethics concerns only"]

**Final Justification:**

I think the author address my concern compared to the SOTA experiments. I hope the authors should add the additional experiments to the new revision. Accordingly, I am raising my score.

**Limitations:**

See questions.

**Paper Formatting Concerns:**

No.

**Quality:**

3

**Strengths And Weaknesses:**

**Strengths**:
1. **Theoretical Innovation**: Proposes Adaptive Fission—the first post-training method enabling dynamic bit-length/weight assignment with guaranteed exponential error reduction.
2. **Hardware Efficiency**: Achieves 80% latency/power reduction via neuromorphic-compatible parallelization while maintaining binary spike compatibility.
3. **Flexible Deployment**: Compatible with ANN2SNN conversion and direct SNN training across classification/generation tasks.

**Weaknesses**:
See questions.

---

> ### Author Rebuttal · Authors · 2025-07-29
>
> We sincerely thank the reviewer for the thorough review and valuable comments. We respectfully hope these clarifications can help resolve your doubts and lead to a more favorable evaluation. Key results will also be incorporated into the final version and appendix if given the opportunity.
>
> ---
>
> ### **Question 1: Computational Cost of Parallel Model Indicators**
> You are absolutely correct that our method introduces exponential growth in comparisons in exchange for finer control of spiking behavior. Despite this, this overall computational cost is still **exponentially more efficient** due to reduced additions, compared to conventional encoding schemes of parallel neurons.
>
> 1. **Complexity analysis on algorithm**
>    For a group of $ n $ split neurons, the parallel model involves $2^{i-1}$ comparisons (CMPs) and $1$ addition (ADD) for the $i$-th neuron, i.e., **$2^n-1$** CMPs and **$n$** ADDs in total, to achieve $2^n$-level precision.
>    In contrast, to match this precision with conventional scheme, $ 2^n-1 $ neurons will be required, each with $1$ CMP and ADD, i.e., **$ 2^n-1 $** CMPs and **$ 2^n-1 $** ADDs in total, where **ADDs increases exponentially** compared to ours.
>    Besides, the cascade version (Fig. 2 in the paper) requires only $ n $ CMPs and ADDs, but its sequential nature limits parallelizability and makes it unsuitable for deployment.
>
> 2. **Hardware cost perspective**
>    Neuromorphic chips often include dedicated CMP circuits due to threshold-based firing mechanisms. Comparators are typically 3–5× more energy-efficient than adders, with smaller delay and hardware footprint. Thus, the CMP-dominant nature of our method aligns well with parallel execution on neuromorphic hardware.
>
> ---
>
> ### **Question 2: Impact of Sensitivity-Based Selection**
>
> Ablations on Stage 1 (sensitivity selection) can lead to the following differences:
>
> 1. **Accuracy**
>    Applying fission to all neurons (without selection) does not hurt performance and may slightly increase maximum accuracy (often $<2\%$), since some rare but important neurons may otherwise be missed. But the added computational and spatial cost outweighs the marginal benefit.
>
> 2. **Redundancy and energy**
>    Without selection, each round of fission adds one neuron per activation value. After $ k $ rounds, the model expands by a factor of $ k+1 $. However, as shown in Figure 6, about 70% of neurons are never fissioned when selection is used. Fission on these inactive units results in negligible gain but increases memory and potential power cost.
>
>    Moreover, in Figure 8, focusing on the rightmost orange bars: full fission (no selection) with 200% additional neuron overhead achieves similar accuracy as sensitivity-based fission with only 60% overhead.
>
> Thus, Stage 1 provides a cost-effective trade-off, concentrating computation on informative neurons without sacrificing accuracy.
>
> ---
>
> ### **Question 3: Comparison with State-of-the-Art Methods**
> We appreciate the reviewer’s suggestion, and have thus extended our evaluation accordingly with the recent SOTA conversion methods mentioned [r1-r4]. We have previously reproduced both SRP [r1] and STA [r4], and now integrate our algorithm to them.
>
> | Method | Time Step | Fission Rate | Accu. (%) |
> |--------|-----------|--------------|-----------|
> | **CIFAR-100 & ResNet20** |
> |SRP($\tau=4$) | 32        | No           | 65.50     |
> |  | 16        | No           | 64.71     |
> |  | 8        | No           | 62.94     |
> |  | 4        | No           | 59.34     |
> | + Our Fission | 16        | 0.28         | **68.72**     |
> |  | 8         | 1.02         | 66.41     |
> |  | 4         | 2.71         | 65.39     |
> | **CIFAR-100 & ViT-B/32(Supervised)** |
> |STA | 64        | No           | **85.25**     |
> |  | 32        | No           | 84.15     |
> | + Our Fission | 16        | 0.53         | 82.70     |
> |  | 8         | 1.15         | 80.28     |
> |  | 4         | 2.35         | 79.14     |
>
> These results confirm that Adaptive Fission provides consistent improvements when applied to SOTA models. Since our method can be regarded as **post-training quantization**, it is broadly compatible with various pre-trained architectures.
>
> - For SRP[r1], which retains an ANN optimization target during conversion, the fissioned model can even **surpass the original SNN baseline** (e.g., at $T=32$) due to finer-grained encoding.
> - STA[r4], as a transformer-based method, is much harder with stricter precision requirements. As a result, more neurons are typically introduced during fission to preserve accuracy, leading to **slightly higher computational cost and accuracy degradation** compared to CNN cases.
>
> For other 2 mentioned SOTA works:
> - **SpikeZIP-TF**[r2], as a quantized-conversion method, requires first retraining a quantized ANN prior to conversion. We are actively extending our method to support this framework.
> - **ECMT**[r3], as a follow-up of [r4], involves multi-threshold neurons. Extending Adaptive Fission to multi-threshold neurons is promising, but beyond the scope of current design assumptions of single-threshold. We are actively exploring this direction.
>
> Besides, due to the deployment complexity of non-CNN models on neuromorphic hardware, real-device benchmarks (latency/power) for transformer based architectures are currently underway and is to be included in future revisions.
>
> &nbsp;
>
> [r1] Reducing ANN-SNN conversion error through residual membrane potential. AAAI 2023.
> [r2] SpikeZIP-TF: Conversion is all you need for transformer-based SNN. ICML 2024.
> [r3] Towards high-performance spiking transformers from ANN to SNN conversion. ACMMM 2024.
> [r4] Spatio-temporal approximation: A training-free SNN conversion for transformers. ICLR 2024.

---

> > ### Comment · Reviewer_W46Z · 2025-08-04
> >
> > I appreciate the authors’ detailed rebuttal, which has resolved my concerns. I hope the authors should add the additional experiments to the new revision. Accordingly, I am raising my score.

---

> > > ### Author Response · Authors · 2025-08-04
> > > **Thanks for your support**
> > >
> > > We sincerely thank the reviewer for the thoughtful comments and for acknowledging our rebuttal. We appreciate your suggestion regarding additional experiments, and we will incorporate them in the final version.

---

### Official Review · Reviewer_hA37 · 2025-07-02

**Clarity:** 4
**Significance:** 3
**Originality:** 4
**Rating:** 5
**Confidence:** 3

**Summary:**

The paper proposes a novel method to improve information coding in spiking neural networks (SNNs), essentially for ANN-to-SNN conversions. Building on current population and encoding existing models, the technical contributions of the paper are as follows: 1) an algorithm to select neurons that require coding improvement, 2) a model that divides selected neurons into subpopulations of neurons and optimizes the neurons’ parameters to improve information coding.

The model was tested on neuromorphic hardware on image classification and image generation tasks and significantly improved energy consumption by reducing processing time, while maintaining competitive performance.

**Questions:**

I would also like to ask the authors to clarify if other existing methods use the cascade / parallel neural grouping models described in Section 4.

Additionally, I also noticed two minor typo: line 26 “needed” -> “need”, and line 43, “yield” -> yielding.

**Ethical Concerns:**

["NO or VERY MINOR ethics concerns only"]

**Final Justification:**

The authors' rebuttal addressed correctly my concerns. I therefore do not decrease my score and maintain it to 5.

**Limitations:**

The authors properly addressed the different limitations of their work in the discussion section.

**Quality:**

4

**Strengths And Weaknesses:**

Strengths:

1) A sound and original model with clear mathematical foundations and which focuses on solving one well identified goal
2) Strong evaluation of the work on multiple state-of-the-art datasets with a report on the accuracy, energy consumption and time processing performances compared to a control case/baseline network
3) Valuable tests on neuromorphic hardware (Lynxi 48 HP201 neuromorphic chip)

Weaknesses:

1) The model requires adding more neurons to the networks to improve information coding. This adds an additional constraint on the memory usage and could be a bottleneck depending on the available resources/the hardware used
2) The model is not compatible with event-driven platforms (e.g. event cameras) where input data vary at each timestep
3) The model requires an additional training phase after conversion of the ANN to SNN or after training from scratch of the SNN

---

> ### Author Rebuttal · Authors · 2025-07-28
>
> We sincerely thank the reviewer for highly positive and encouraging feedback on our work. We hope our responses can help further clarify the contributions and limitations of our paper. If accepted, we will correct the minor issues you pointed out in the final version.
>
> ---
>
> ### **Weaknesses**
> We acknowledge the three limitations raised in the Strengths and Weaknesses section. To some extent, these are inherent trade-offs of our encoding strategy for inference acceleration. Since post-training techniques are often considered a "no free lunch" solution, our goal is to minimize the overhead while achieving substantial latency and energy improvements. Specifically,
>
> 1. **Memory Overhead:**
>    Adaptive Fission only selectively and sparsely introduces additional neurons to a small subset of high-sensitivity neurons, avoiding the redundancy caused by uniformly allocating bit-widths to all activations in conventional quantization.
>
> 2. **Event-driven Compatibility:**
>    We agree that the current method is more suitable for static inputs (e.g., images) rather than event-driven data streams such as event cameras. This is explicitly acknowledged in our discussion section, and is to be improved in future work.
>
> 3. **Post-training Adjustment:**
>    Regarding the additional training phase after conversion, we consider this to be more of a calibration step [28, r1, r2] rather than training, retraining or finetuning. This practice is also common in post-training quantization for ANNs [r3], and has two key properties:
>    - **No weight update** is performed; instead, it adjusts a few parameters based on statistical properties extracted from a small calibration set, thus avoiding feature distortion or catastrophic forgetting.
>    - It requires **very few samples** and typically no multi-epoch optimization, resulting in negligible overhead.
>
>     Hence, we believe this lightweight step does not impose significant practical cost.
>
> &nbsp;
>
> [r1] Error-aware conversion from ANN to SNN via post-training parameter calibration. IJCV 2024.
> [r2] Adaptive calibration: A unified conversion framework of spiking neural networks. AAAI 2025
> [r3] Accurate post training quantization with small calibration sets. ICML 2021.
>
> ---
>
> ### **Questions: Similar models in related works**
> To the best of our knowledge, no prior work adopts the same form of "cascade or parallel neuron grouping" proposed in this paper. Some prior methods [24, 32] do involve neuron grouping but significantly differ in two key aspects:
>
> - **Limited representational power:**
>   For instance, [24] uses multiple parallel neurons with uniformly divided thresholds to approximate multi-level firing. The $k$-th neuron is assigned a threshold scaled by $k/n$, and only one spike can be emitted at a time — yielding **linear** rather than **exponential** precision improvements.
>   Similarly, [32] employs small auxiliary networks, but still lacks highly-independent binary combinatorial capability like our $\mathbb{I}_i$ indicators (Eq. 15), again resulting in only linear improvements.
>
> - **Fixed weight allocation:**
>   The combinations in prior works are either **uniformly preset** [24] or **learned via separate pre-training** [32], rather than dynamically determined based on neuron sensitivity as in our method.

---

> > ### Comment · Reviewer_hA37 · 2025-08-07
> >
> > I thank the authors for their responses. They address well the concerns I had and I will therefore maintain my current score.

---

### Official Review · Reviewer_r9t1 · 2025-07-03

**Clarity:** 2
**Significance:** 3
**Originality:** 3
**Rating:** 4
**Confidence:** 3

**Summary:**

This manuscript introduces Adaptive Fission, a post-training population coding technique designed to reduce latency and power consumption in Spiking Neural Networks (SNNs) without sacrificing accuracy. Experimental results show significant improvements in performance and energy efficiency.

**Questions:**

What is the intuition or intuitive idea behind the adaptive fission proposed here?

**Ethical Concerns:**

["NO or VERY MINOR ethics concerns only"]

**Limitations:**

The limitation is only briefly discussed regarding the structural complexity incurred by the proposed technique.

**Paper Formatting Concerns:**

None found.

**Quality:**

3

**Strengths And Weaknesses:**

The main advantage is that this paper offers strong theoretical foundation and theorems supporting its design. The main disadvantage is that some of the theoretical derivations are hard to follow.

Here are some detailed comments,
1,There could be more explanation for the notations and derivative of some equations, such as equations (5) and (6). Besides, citations are needed here.
2, There are dense notations in equation (15), (16) and (21), which make them hard to be understood; some intuitive explanation could help here. Also the number of the equation between (20) and (21) is missing.
3, The discussion of sensitivity-based selection strategy can be expanded with its limitations.
4. Some minor grammatical errors.

---

> ### Author Rebuttal · Authors · 2025-07-28
>
> We sincerely thank the reviewer for detailed and constructive feedback. We address the raised concerns below and will incorporate the relevant clarifications into the revised version if possible.
>
> ---
>
> ### **Weakness 1: Clarification of conversion techniques in Equations (5) and (6)**
>
> Thank you for pointing this out. Equations (5) and (6) follow established formulations in ANN-to-SNN conversion literature.
>
> - **Eq.(5)** is similar to scaling techniques in training-free SNN conversion, such as [28, Eq.5] and [45, Eq.5], which address the mismatch  between spike outputs in $\lbrace 0, V_{th}\rbrace$ model and normalized outputs in standard $\lbrace 0,1\rbrace$ IF model. For example, in [28]:
> \begin{equation}
> \mathbf{W}^{(\ell)} \leftarrow \frac{V_{th}^{(\ell-1)}}{V_{th}^{(\ell)}} \mathbf{W}^{(\ell)}, \quad V_{th}^{(\ell)} \leftarrow 1
> \end{equation}
> *To illustrate: the previous layer firing is either $0$ or $V_{th}^{(\ell-1)}$, and passes through a synapse with weight $\mathbf{W}^{(\ell)}$. The neuron receives $V_{th}^{(\ell-1)} \times \mathbf{W}^{(\ell)}$ as input and compares it to threshold $V_{th}^{(\ell)}$ to decide firing. This comparison is equivalent to checking whether $\frac{V_{th}^{(\ell-1)}}{V_{th}^{(\ell)}} \mathbf{W}^{(\ell)} > 1$, allowing us to absorb the threshold ratio into the weights and fix the threshold at 1.*
>
>   Similarly, in our **Eq.(5)**, since the spike output $s(t)$ in Eq.(3) lies in $\lbrace  0, \theta \rbrace$, Eq.(5) linearly rescales $w$ and $s(t)$ via $w := \theta w$ and $s(t) := s(t)/\theta$, so that the computation is compatible with neuromorphic hardware supporting $\lbrace 0,1\rbrace$ spikes.
> &nbsp;
> - **Eq.(6)** is a technique commonly used in quantization-based conversion methods (e.g., [28, Eq.9], [7, Eq.7]), reflecting the mapping between SNN spike counts and quantized ReLU outputs. This can be interpreted as approximating the discrete nature of SNN outputs ($\lbrace 0, 1, ..., T\rbrace$) via clipping and rounding operations in ReLU ANN. Its derivation can be found in the relevant citations.
>
> ---
>
> ### **Weakness 2: Intuitive explanation of Equations (15), (16), and (21)**
>
> We appreciate your observation and agree these parts can benefit from improved intuition.
>
> - **Eq.(15)** describes how our neuron group transitions from a cascading to a parallel form. In Definition 4.5, we use a cascade-style explanation: higher-threshold neurons fire first, and any residual membrane potential is passed to lower-threshold neurons for further comparison. While conceptually clear, this mechanism is inefficient in practice.
> Instead, we observe that the firing judgment for lower neurons can be **precompiled based on the total input, without waiting for upstream neuron activity**. For example, if two neurons have thresholds 2 and 1, the lower neuron can directly determine firing when the input falls in $[1,2) \cup [3, \infty)$, turing it into lightweight comparison logic. These firing rules are represented by our proposed indicator function.
> &nbsp;
>
> - **Eq.(16)** estimates each neuron’s output sensitivity to residual error. It uses forward propagation to estimate **average residual potential** for each neuron, and approximated back propagation to derive **gradients of the output** relative to residual potential. Their product serves as a first-order approximation of how much residual error affects the final output.
> &nbsp;
>
> - **Eq.(21)** gives a high-level summary of our optimal threshold derivation. The main text only presents a simplified form, and we refer the reviewer to Appendix A.2 for a complete proof.
>
>   In brief, the error function with respect to threshold $\theta_k$ is step-wise and highly discontinuous. Even a small perturbation in $\theta_k$ may lead to a large change in approximation error, making direct optimization of $\theta_k$ extremely unstable.
>
>   To address this, we instead **represent the error with the cumulative distribution function $F$** of the original floating-point activation values. This function possesses more desirable properties — with sufficient sampling, $F$ is approximately continuous and differentiable. This enables us to estimate gradients with respect to $\theta_k$ and derive near-optimal thresholds.
>
> ---
>
> ### **Weakness 3: Limitations of sensitivity-based strategy**
> The inherit main challenge with the sensitivity-based selection arises when calibration samples are limited, leading to unreliable estimates. This issue has also been discussed in early ANN quantization works such as [r1-r3]. All these works emphasize the need for layer/channel-specific treatment due to varying sensitivity to quantization.
>
> Nevertheless, their sensitivity analysis components are often relatively simple — typically relying on coarse statistics such as activation magnitude or gradient norms. This leaves room for further exploration of more fine-grained or theoretically grounded sensitivity estimation methods, particularly in the context of post-training adaptation.
>
> &nbsp;
>
> [r1] Quantization and training of neural networks for efficient integer-arithmetic-only inference. CVPR 2018.
> [r2] Improving neural network quantization without retraining using outlier channel splitting. ICML 2019.
> [r3] Smoothquant: Accurate and efficient post-training quantization for large language models. ICML 2023.
>
> ---
>
> ### **Question: The intuition**
>
> Our approach is motivated by the connection between SNNs and quantized ANNs, both restricted by insufficient precision. We observe that ANN quantization typically imposes two rigid structural constraints:
> (1) a **fixed bitwidth** across all weights and activations (e.g., W4A4 assigns 4 bits uniformly),
> (2) **fixed bitwise weight combinations** (e.g., binary weights with powers-of-two significance $1,2,4,\dots$),
> which are not inherently necessary in SNNs.
>
> This motivates us to design an allocation scheme that allows both adaptive **bitwidth** and **bitwise scaling factors**, depending on neuron-level requirements. Since searching for a globally optimal bit allocation is combinatorially hard, we instead propose a sparse and iterative local strategy that incrementally allocates additional “bits” (i.e., spiking sub-neurons) only where needed. This insight forms the core of our adaptive fission mechanism.
>
> ---
>
> ### **Limitations: Discussion on structural complexity**
>
> We discuss structural overhead in Appendix C and in response to Reviewer QkMA (Q2). As deployment overhead may depend on the hardware backend, we provide both algorithm-level spatial complexity analysis and real-world hardware measurements. We are open to provide further details on case studies.

---

> > ### Comment · Reviewer_r9t1 · 2025-08-06
> > **Interesting work**
> >
> > The authors' clarifications are well received and appreciated. One more minor point:
> >
> > Refs. [36] and [37] are the same.

---

### Note · Authors · 2025-08-12

We thank all reviewers and the AC for their constructive and encouraging feedback. We appreciate the positive assessment of our work, and if given the opportunity, we will incorporate the suggested minor clarifications and additional results into the final version.

---

### Decision · Program_Chairs · 2025-09-17

**Decision:**

Accept (poster)

**Comment:**

This paper proposes a post-training encoding method that converts ANNs to SNNs, substantially reducing the number of SNN inference steps. Initial reviews were mixed, but after the rebuttal all reviewers recommended acceptance. The reviewers highlighted the method's solid theoretical foundation and strong empirical results, particularly the evaluation on real hardware. The AC concurs with the reviewers' post-rebuttal assessment and recommend acceptance. The AC encourages the authors to incorporate the rebuttal materials into the camera-ready version.